# NEEDLEINATABLE: Exploring Long-Context Capability of Large Language Models towards Long-Structured Tables

**Lanrui Wang**[1][*][†], **Mingyu Zheng**[1,2][*], **Hongyin Tang**[3][*][†], **Zheng Lin**[1,2][‡],
**Yanan Cao**[1,2], **Jingang Wang**[3][†], **Xunliang Cai**[3][†], **Weiping Wang**[1]
[1]Institute of Information Engineering, Chinese Academy of Sciences
[2]School of Cyber Security, University of Chinese Academy of Sciences
[3]Meituan
oliveerwang@tencent.com
{zhengmingyu,linzheng}@iie.ac.cn    tanghongyin@meituan.com

## Abstract

Processing structured tabular data, particularly large and lengthy tables, constitutes a fundamental yet challenging task for large language models (LLMs). However, existing long-context benchmarks like Needle-in-a-Haystack primarily focus on unstructured text, neglecting the challenge of diverse structured tables. Meanwhile, previous tabular benchmarks mainly consider downstream tasks that require high-level reasoning abilities, and overlook models' underlying fine-grained perception of individual table cells, which is crucial for practical and robust LLM-based table applications. To address this gap, we introduce NEEDLEINATABLE (NIAT), a new long-context tabular benchmark that treats each table cell as a "needle" and requires models to extract the target cell based on cell locations or lookup questions. Our comprehensive evaluation of various LLMs and multimodal LLMs reveals a substantial performance gap between popular downstream tabular tasks and the simpler NIAT task, suggesting that they may rely on dataset-specific correlations or shortcuts to obtain better benchmark results but lack truly robust long-context understanding towards structured tables. Furthermore, we demonstrate that using synthesized NIAT training data can effectively improve performance on both NIAT task and downstream tabular tasks, which validates the importance of NIAT capability for LLMs' genuine table understanding ability. [4].

## 1 Introduction

The long-context modeling ability has laid the foundation for a wide range of LLM-based applications, such as lifelong conversational chatbots, document analysis tools, and advanced agent-based systems [1–4]. Correspondingly, lots of effort has been dedicated to scaling up the context window of LLMs, improving their proficiency in handling extremely long input texts [5–9]. In parallel, various benchmarks like Needle-in-a-Haystack (NIAH) [10] have been developed to evaluate the long-context capabilities of LLMs from diverse perspectives, such as extracting key information from long texts, answering questions about extended passages or multiple documents [11–15].

---

[*]   Equal contribution. Work was done when Lanrui Wang was a graduate student of IIE, CAS.
[†]   Joint work with Meituan
[‡]   Zheng Lin is the corresponding author.
[4]Our code and data are available at: `https://github.com/wlr737/NeedleInATable`

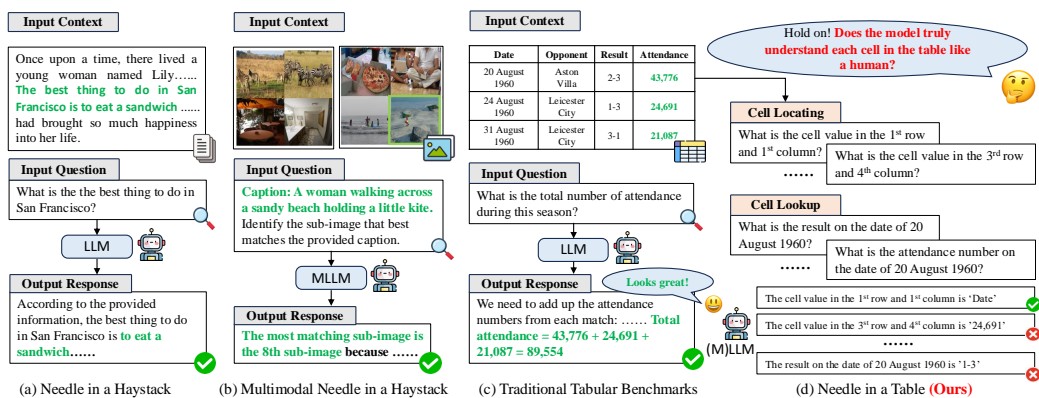

Figure 1: Comparison of previous long-context benchmarks, tabular benchmarks and the proposed NIAT benchmark. Existing long-context benchmarks overlook the structured tabular data, while traditional tabular benchmarks mainly focus on high-level complex reasoning ability. Both of them ignore the model's basic fine-grained comprehension of individual cells in the table context.

Beyond lengthy unstructured textual data, structured tables also play a critical role in real-world long-context scenarios. Featured with unique two-dimensional structures and diverse formats, tables are widely used in different domains to organize and present information, ranging from financial documents, scientific research to medical records [16–18]. Moreover, LLM-based chatbot users often feed serialized tables in different formats like Markdown into LLMs' context windows to perform various table understanding tasks such as table question answering (TQA) and table fact verification (TFV) [19, 20]. Consequently, the lengths of these table-related task requests often exceed that of common instructions with a few sentences, presenting a distinct long-text challenge for LLMs.

However, existing long-context benchmarks primarily focus on evaluating the ability of LLMs to comprehend long unstructured text, overlooking in-depth exploration of long-context scenarios of structured tables. On the other hand, previous tabular LLM studies mainly assess LLMs' proficiency with traditional benchmarks of downstream tabular tasks that demand high-level reasoning abilities based on partial table information, but neglect the importance of pressure testing LLMs' underlying fine-grained understanding of each individual table cell. This is crucial for practical LLM-based table applications as real-world users could input questions about any table cells. Besides, it can help determine whether we have developed a truly robust table understanding ability or if we are barking up the wrong tree by overly pursuing the model performance on specific benchmarks.

To bridge this gap, we introduce a new task and construct a benchmark based on publicly available tables, termed NEEDLEINATABLE (NIAT). As illustrated in Figure 1, unlike previous long-context benchmarks and tabular benchmarks, the proposed NIAT task treats each table cell as a "needle" and requires the model to extract the target cell according to two types of questions, respectively. The cell-locating questions ask the model to locate the specific cell at the given position, and the cell-lookup questions require the model to retrieve the answer cell based on the information lookup demand. Our benchmark contains 750 tables and up to 287K questions in total, covering three table structures (flat, horizontal and hierarchical tables), three table formats (Markdown, HTML and image) and diverse table sizes (fixed and arbitrary).

With the help of NIAT benchmark, we evaluate a wide spectrum of LLMs and MLLMs including mainstream open-source models and close-sourced GPT-4o. These models demonstrate varying performance on the NIAT task and they obtain much better results on cell-lookup questions than cell-locating questions. This shows that current LLMs can retrieve answer cells from input tables using the attention mechanism, but they struggle in interpreting the basic two-dimensional table structures in a human-like perspective. More importantly, our thorough evaluation reveals a substantial performance gap between more complex downstream tabular tasks (like TQA and TFV) and the NIAT task, suggesting that they may rely on dataset-specific correlations or shortcuts to obtain better benchmark results but lack truly robust long-context understanding towards structured tabular data.

Furthermore, we explore whether enhancing NIAT capabilities can fundamentally improve the overall table understanding ability. To this end, we propose a strong2weak data synthesis method to create NIAT fine-tuning data. Specifically, we first utilize the in-context learning with GPT-4o to generate various NIAT task queries based on tables from training splits, and then synthesize detailed chain-of-thought (CoT) reasoning processes as target responses, which instruct weak LLMs to progressively understand structured tabular data. Experimental results with Llama3.1-8B-Instruct and Qwen2.5-7B-Instruct demonstrate that our synthesized 12K NIAT training data can not only improve models' long-context performance on the NIAT task, but also achieve a significant performance gain (14.55%↑ in average accuracy) on four downstream benchmarks (WTQ, TabFact, HiTab and TABMWP), outperforming strong baselines such as long-context LLMs and specialist tabular LLMs, which further underscores the importance of NIAT capabilities. We hope that this work could foster advancements in both long-context LLM and tabular LLM communities, facilitating the development of LLMs with robust understanding of complex and lengthy structured tables.

We conclude our contributions as follows:

- Targeted at the limitations of existing benchmarks for long-context and table understanding, we introduce a new task named NIAT and construct the first benchmark for evaluating LLMs' long-context table understanding ability.
- On this basis, we conduct a thorough evaluation of various types of LLMs and MLLMs, revealing their shortcomings in long-context understanding of lengthy tables and providing valuable insights.
- We propose a simple yet effective data synthesis method to enhance LLMs' long-table comprehension capabilities, which also significantly improves their performance on downstream table-related tasks, outperforming recent state-of-the-art baselines.

## 2 Related Work

**Long Context LLMs and Benchmarks.** Recently, the long-context capabilities of LLMs have garnered significant attention in the research community. Numerous efforts have focused on enhancing positional embeddings to develop more powerful long-context LLMs [21–24]. Extensive studies have also explored and evaluated LLMs' ability to understand long-context data across various modalities [25, 26, 13] and generate long-form content [27, 28]. To evaluate the long-context capabilities of large language models (LLMs) in the text modality, benchmarks such as RULER [15], Infini-Bench [11], and LongBench [1] have introduced diverse long-context understanding tasks across varying context lengths, providing a comprehensive evaluation framework for developing robust long-context LLMs. However, these benchmarks primarily focus on unstructured text, largely overlooking structured table data. In this paper, we address this gap by proposing the NIAT tasks.

Notably, a concurrent long-context benchmark, LongBench-v2 [12], evaluates the high-level reasoning ability of LLMs over long structured tables using 18 question-answering samples. In contrast, our work focuses on assessing LLMs' comprehension of every individual cell within the global context of long tables, making it complementary to LongBench-v2.

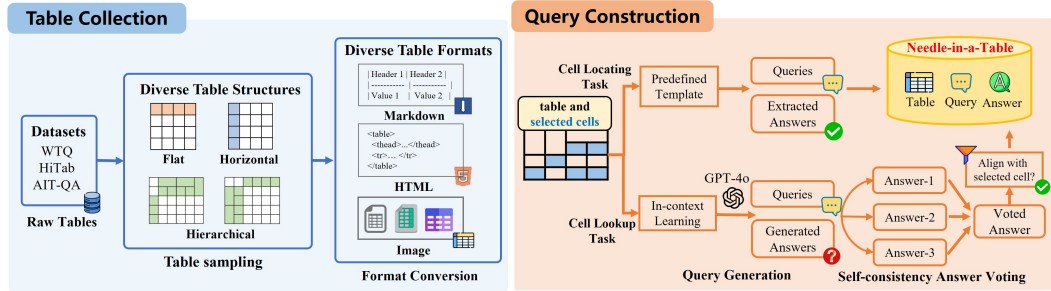

Figure 2: The construction pipeline of Needle-in-a-Table Benchmark.

**Tabular LLMs and Benchmarks.** Recent years have witnessed a paradigm shift to developing Tabular LLMs for table understanding, where different strategies have been applied to enhance LLMs'

ability to fulfill tabular tasks, such as prompt engineering [29, 30] and instruction tuning [31–34]. Moreover, a series of benchmarks have been constructed to evaluate LLMs' proficiency in table understanding tasks within diverse scenarios [35–37]. Nonetheless, these benchmarks primarily focus on assessing LLMs' high-level table-based reasoning ability with complex tasks as testbeds, such as table question answering and table-based fact verification, and fail to evaluate the fine-grained understanding of each input table cell from the long-context perspective, which is the foundation for humans to comprehend tables and perform more complex tasks. To this end, we propose the NIAT benchmark to address this issue and provide a comprehensive landscape of LLMs' table understanding capacity over the whole input table.

## 3 NEEDLEINATABLE Benchmark

### 3.1 Benchmark Construction

#### 3.1.1 Table Collection

To fill the gap in existing benchmarks and provide the community with the first long-context benchmark towards structured tables, it is most cost-effective to build such a benchmark based on tables from publicly aviable datasets. Moreover, real-world tables possess various structures, ranging from simple flat tables to hierarchical tables with multi-level headers and merged cells [16, 38, 39]. With these considerations in mind, we collect tables of three different structures from existing tabular benchmarks. **(1) Flat tables**: we randomly select 250 tables from WTQ benchmark [19] which contains flat tables from Wikipedia. **(2) Hierarchical tables**: we randomly select 220 tables and 30 tables from HiTab [40] and AIT-QA [41] benchmarks respectively, which contain diverse hierarchical tables from Wikipedia, statistic reports, and annual reports of airline companies. **(3) Horizontal tables**: we write Python scripts to transpose flat tables in WTQ so that the first column serves as row headers, resulting in 250 horizontal tables.

Another unique feature of structured tables lies in the diversity of table representation formats [30, 42, 34]. Chatbot users may input tables in different formats into LLMs' context windows, such as Markdown, HTML, csv, Latex, and even image format. Therefore, we carefully design Python scripts to transform the collected tables into three commonly used formats including **Markdown, HTML and image** formats. Table sequences in Markdown and HTML formats are used to evaluate textual LLMs. For image format, we render tabular data into images following the previous work [34], and the resulting table images serve as inputs for multimodal LLMs to evalute their NIAT capabilities.

#### 3.1.2 Query Construction

Despite the remarkable performance of LLMs on complex tabular benchmarks, even surpassing human levels, does this really represent that LLMs possess the same robust long-context table understanding abilities as humans? To answer this question, we take a step back to the fundamental or atomic ability of humuan-like table comprehension, focusing on fine-grained perception of individual table cells. Concretely, we treat the input table as the context, the single table cell as the target "needle", and design two types of NIAT queries that are simple and trivial for humans but constitute the foundation of more complex table-based tasks.

**Cell Locating** queries require the model to extract the cell content according to the given cell location (i.e., row index and column index), which aims at evaluating the understanding of basic two-dimensional table structures. An example query could be "What is the cell value in the 3rd row and 4th column". If the model has a correct understanding of the basic table structure, it should be easy to locate the target cell by enumerating along the row and column directions.
**Cell Lookup** queries are simple lookup questions where answers are specific table cells and do not need any further aggregate operations (such as min/max/sum/count) or multi-step reasoning. Using the table in Figure 1 as an example, multiple lookup questions can be raised based on different table cells, e.g., "What is the result on 20 August 1960". This task requires the model to use the keywords in the questions as cues and retrieve the answer cell from the table by intersecting the row and the column with overlapped keywords.

To save the cost of model inference, we randomly select 60% table cells from each table as target cells to create NIAT queries. Since the cell locating task only takes the table and the target cell location as input, we fill in the pre-defined prompt template with selected row indices and column

Table 1: Performance of LLMs on NIAT benchmark. 'Flat', 'Hori.' and 'Hie.' denote flat, horizontal and hierarchical tables, respectively. 'MD.' and 'HT.' indicates tables in Markdown and HTML formats. The best results in each category are highlighted in **bold**.

| Model | Cell-Locating | | | | | | | Cell-Lookup | | | | | | | Over all |
|---|---|---|---|---|---|---|---|---|---|---|---|---|---|---|---|
| | Flat | | Hor. | | Hie. | | Ave | Flat | | Hor. | | Hie. | | Ave | |
| | MD. | HT. | MD. | HT. | MD. | HT. | | MD. | HT. | MD. | HT. | MD. | HT. | | |
| *Open-source LLMs* | | | | | | | | | | | | | | | |
| Mistral-7B-Instruct-v0.3 | 7.75 | 6.79 | 4.35 | 6.33 | 2.11 | 0.28 | 4.60 | 33.72 | 38.79 | 27.99 | 27.55 | 25.62 | 33.79 | 31.98 | 18.29 |
| Deepseek-llm-7B | 2.88 | 0.16 | 2.31 | 2.03 | 1.13 | 0.70 | 1.54 | 47.46 | 19.37 | 32.21 | 27.03 | 38.29 | 26.05 | 31.74 | 16.64 |
| MiniCPM-3-4B | 1.48 | 2.01 | 0.87 | 1.57 | 0.73 | 0.40 | 1.18 | 64.60 | 66.51 | 49.65 | 51.54 | 59.12 | 57.38 | 58.13 | 29.66 |
| InternLM2.5-7B-chat | 1.36 | 1.36 | 1.05 | 1.34 | 0.54 | 0.09 | 0.96 | 36.31 | 37.34 | 27.85 | 27.03 | 43.15 | 39.89 | 35.26 | 18.11 |
| Yi-1.5-9B-chat-16K | 7.29 | 5.36 | 4.31 | 7.45 | 1.77 | 0.26 | 4.41 | 52.81 | 45.26 | 43.14 | 44.26 | 54.31 | 54.52 | 49.05 | 26.73 |
| GLM-4-9B-chat | 8.33 | 9.76 | 5.76 | 7.32 | 2.45 | **6.83** | 6.74 | 55.09 | 48.48 | 42.40 | 36.89 | 61.20 | 50.98 | 49.17 | 27.96 |
| Qwen2.5-7B-Instruct | 16.24 | 13.64 | 7.63 | 15.85 | 2.90 | 0.49 | 9.46 | 44.80 | 48.20 | 38.92 | 42.39 | 56.40 | 54.90 | 47.60 | 28.53 |
| Qwen2.5-14B-Instruct | 19.18 | 18.28 | 11.41 | 26.27 | 4.69 | 0.39 | 13.37 | 42.73 | 36.80 | 38.29 | 36.46 | 56.16 | 51.84 | 43.71 | 28.54 |
| Llama3.1-8B-Instruct | 10.38 | 8.75 | 7.18 | 7.70 | 2.54 | 0.38 | 6.16 | 67.30 | 65.10 | 57.10 | 66.25 | 72.50 | 66.20 | 65.74 | 35.95 |
| Qwen3-14B | 17.6 | 18.73 | 18.41 | 23.81 | 4.33 | 0.81 | 13.95 | 74.41 | 70.76 | 64.14 | 62.2 | 72.79 | 79.59 | 70.65 | 42.30 |
| Qwen3-32B | 22.41 | **22.6** | 21.32 | **31.05** | **5.77** | 1.19 | **17.39** | 76.6 | 71.24 | 69.23 | 63.8 | 80 | 80.6 | 73.58 | 45.48 |
| Qwen3-30B-A3B | **23.17** | 20.41 | **24.12** | 26.2 | 3.65 | 1.39 | 16.49 | **78.87** | **78.83** | 72.2 | 69.8 | 85.41 | 86.59 | 78.62 | 47.55 |
| *Tabular LLMs* | | | | | | | | | | | | | | | |
| StructLLM | 2.00 | 2.10 | 1.30 | 4.20 | 1.20 | 0.10 | 1.82 | 58.65 | 46.70 | 25.85 | 29.65 | 51.85 | 43.55 | 42.72 | 22.27 |
| TableGPT2 | **12.67** | **12.83** | **6.32** | **17.52** | **3.14** | **0.52** | **8.84** | **77.02** | **71.31** | **61.03** | **71.64** | **80.73** | **81.50** | **73.87** | **41.36** |
| *Reasoning LLMs and GPT-4o* | | | | | | | | | | | | | | | |
| GPT-4o | 38.50 | 29.70 | 36.50 | 39.40 | 10.40 | 1.50 | 26.00 | 61.50 | 63.80 | 60.30 | 59.50 | 83.60 | 81.10 | 68.30 | 47.15 |
| Qwen-QwQ-32B | 63.76 | 54.40 | 36.36 | 52.29 | 3.90 | 1.80 | 35.42 | 48.78 | 45.21 | 45.14 | 45.84 | 56.55 | 56.90 | 49.74 | 42.58 |
| DeepSeek- R1 | **76.20** | **79.50** | **75.46** | **82.30** | **72.39** | **9.60** | **65.91** | **81.00** | **77.82** | **79.19** | **77.50** | **85.60** | **84.80** | **80.99** | **73.45** |

indices to create final input queries. For cell lookup task, we provide GPT-4o with the table and the selected target cell, and utilize in-context learning with GPT-4o to generate corresponding lookup questions. We add JSON output requirements in prompts of two tasks to minimize errors during answer parsing. To guarantee the validity of synthesized cell lookup questions, we use GPT-4o with self-consistency [43] to answer these questions and filter out questions whose answers do not align with the target cells. Finally, we obtain 142K cell-lookup queries and 145K cell-locating queries.

## 3.2 Benchmark Statistics

We have developed NEEDLEINATABLE comprising 750 table and 287K test cases, with an average of 382.67 test cases per table. 80% tables have a row size ranging from 1 to 40, and a column size spaning from 1 to 23. 80% test data have a prompt length within 4.2K, with the maximum length up to 120K. For tables in image format, we randomly select 120 tables from each structure and their corresponding queries as test data for multimodal LLMs. The key statistics about collected tables and constructed queries are shown in Tables 5 and 6, such as average row number and column number, average input length, and so on. Traditional tabular benchmarks mainly evaluate models' complex tabular reasoning with limited test cases, which can only span a small number of table cells. By contrast, the NIAT benchmark evaluates the underlying fine-grained perception of individual table cells with much more test cases, covering various table structures, formats and sizes. **Detailed information about NIAT benchmark are shown in Appendix B.**

## 3.3 Evaluation and Analysis

**Baselines.** We evaluate models across four categories: (1) *Generalist LLMs and MLLMs*, such as Mistral-7B-Instruct-v0.3 [44], Llama3.1-8B-Instruct [8], and InternVL-2.5-8B [45]. (2) *Tabular LLMs and MLLMs* that are fine-tuned with table instruction-tuning data, including StructLLM [46], TableGPT2 [33] and Table-LLaVA [34]. (3) *Recent Reasoning LLMs* including DeepSeek-R1 [47], QwQ-32B [48] with enhanced reasoning ability derived from reinforcement learning. (4) *Close-sourced GPT-4o.* Considering the high cost of GPT-4o and DeepSeek-R1 API, for tables in Markdown,

HTML and image formats, we sample 3K test samples of two types of NIAT tasks, respectively. For all baselines, we adopt the zero-shot setting during evaluation.

**Performance of LLMs.** From the results in Table 1, we have the following findings. (1) Different open-source LLMs demonstrate varying performance on NIAT tasks, with Qwen2.5 family performing best on cell-locating task and Llama3.1-8B-Instruct on cell-lookup task. The competitive performance of Qwen2.5 series models on both NIAT tasks could be stemmed from the specially collected table-related post-training data for enhancing table understanding capability [49]. MiniCPM3-4B also obtains strong performance on cell-lookup task and even greatly surpasses some LLMs with larger sizes, showcasing the potential for developing efficient LLMs with great table understanding ability.

(2) Existing LLMs and tabular LLMs demonstrate much better performance on cell-lookup task compared to cell-locating task. This suggests that current LLMs can utilize semantic co-occurrence between the question and the table to identify question-related rows and columns, and subsequently extract the answer cells via row-column intersections. However, they struggle in accurately comprehending the basic table structures, indicating the fundamental gap between the ways of LLMs and humans in understanding tabular data. To shed more light on the LLMs' perspective of structured tables, we further analyze the attention weights distribution from representative LLMs.

As illustrated in Fig. 3, we have observed certain attention patterns. This figure visualizes the attention weights, where the input table is serialized into a sequence in a left-to-right, top-to-bottom order. The horizontal and vertical coordinates (m, n) correspond to the cell in the m-th row and n-th column, and the color intensity represents the attention weight—brighter colors indicate larger weights. Based on this, we categorized the observed patterns into the **Multi-Slash** and **Local-Triangle** patterns. In the *Multi-Slash* pattern, the attention weights are concentrated on the cell tokens in the same column, thereby exhibiting multiple slash lines at fixed intervals (i.e., the number of cells in one row). In the *Local-Triangle* pattern, the attention weights are concentrated on the cell tokens in the same row within local windows, especially the row header. With these attention patterns, LLMs could retrieve the target cell located in the row and the column that contain keywords of cell-lookup questions. Nevertheless, such a table understanding approach may still fail to guarantee the robust interpretation of table structures, leading to failed edge cases that restrict the reliability of LLM-based applications.

(3) Results in Table 1 on tables of arbitrary sizes show that seemingly powerful LLMs actually lack robust long-context table understanding ability, especially table structure comprehension. To more intuitively demonstrate the influence of different table sizes and increased context length, we crop flat Markdown tables into fixed sizes (15 tables for each size), ranging from 8x8, 12x12, to 32x32, and evaluate the ability of representative LLMs to extract cells from each table position with newly constructed 48K cell-locating queries.

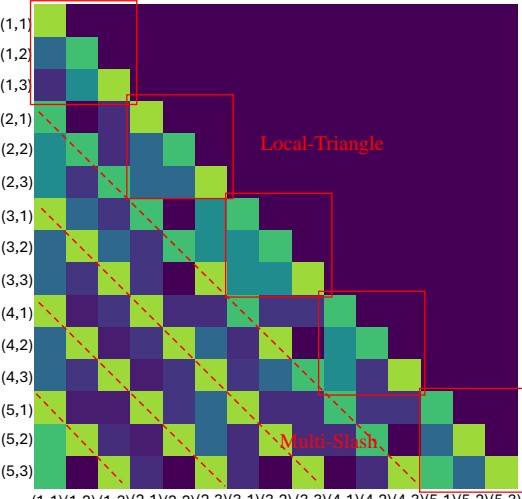

From the per-cell accuracy heatmap illustrated in Figure 4, we can find that GPT-4o demonstrate much better table structure understanding capacity than open-source LLMs like Qwen-2.5-7B-Instruct and Llama3.1-8B-Instruct, with more green color in the heatmap. More importantly, we observe that there exists **lost-in-the-middle-table** phenomenon for LLMs including GPT-4o in understanding lengthy tables [50], i.e., LLMs possess better perception of table cells in the first row and the last row, but struggle with cells in the middle part of tables. Besides,

Figure 3: The illustration of LLMs' attention patterns for structured tables. The input tables are in Markdown format and '(m,n)' indicates cell tokens in the *m-th* row and *n-th* column.

as the table size increases, LLMs suffer a significant performance decline, highlighting the unique challenge of lengthy tabular data to LLMs' long-context abilities. Unlike traditional 'Needle-in-a-haystack' where current LLMs have achieved near-perfect performance, there remains significant room for improvement in the NIAT task.

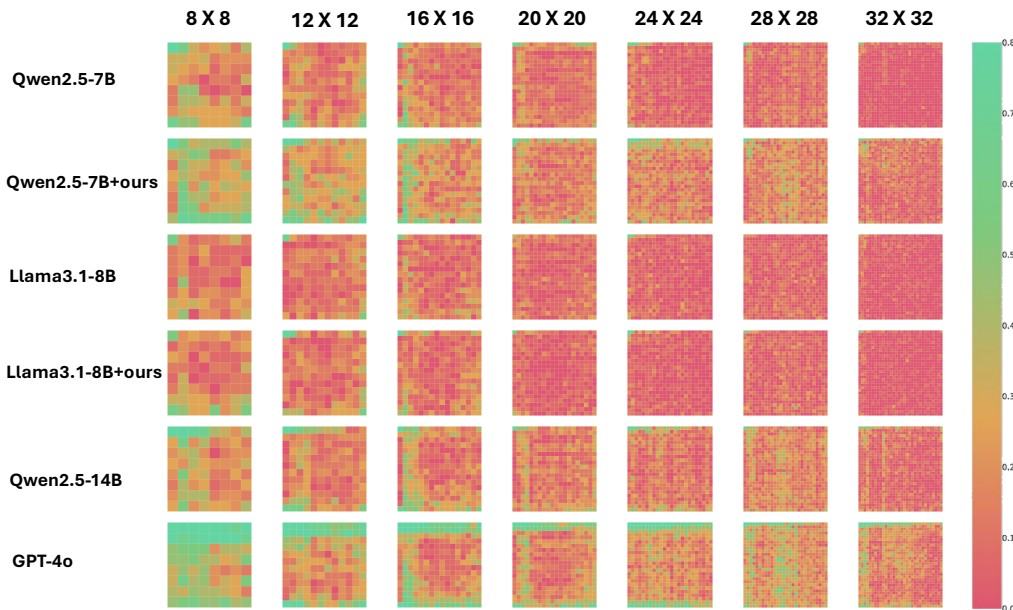

Figure 4: The per-cell accuracy heat map of cell-locating task on tables of fixed sizes, with redder indicating lower accuracy and greener indicating higher accuracy. The x-axis represents tables of different sizes (e.g., 8 × 8 denotes tables with 8 rows and 8 columns), while the y-axis lists evaluated LLMs in instruct version. + *ours* denotes models fine-tuned with our synthesized NIAT training data.

(4) The performance of hierarchical tables is relatively worse than the other two table structures, demonstrating the increased difficulty due to multi-level headers and merged cells. Compared with Markdown format, most models obtain worse results with HTML format, showcasing the unrobustness of current LLMs in handling various table formats and structures, which is consistent with findings from prior works [42, 30].

(5) Compared to generalist LLMs, recent tabular LLM such as TableGPT2 achieve better results, validating the effectiveness of its 2.36M in-house table instruction tuning data. Among various baselines, DeepSeek-R1 achieves the state-of-the-art performance across different settings, demonstrating that test-time scaling techniques could effectively enhance LLMs' ability to process long-structured tabular data. By analyzing model outputs, we find that its in-depth reasoning capability enables more accurate understanding of table structure and extraction of target cells, thereby facilitating more complex downstream tasks.

**Performance of MLLMs.** MLLMs directly take table images as input, which more closely mimic the way humans understand tabular data. From the results in Table 2, we have the following observations. (1) Recent state-of-the-art MLLMs like Skywork-R1V3-38B achieve superior performance in NIAT tasks compared to earlier models, demonstrating the advantages of their optimizations in understanding larger table images, such as supporting dynamic high image resolution and introducing extra post-training data for text-intensive images.

(2) Similar to the performance trend observed in LLMs, multimodal LLMs also perform significantly worse on cell-locating tasks compared to cell-lookup tasks, which reveals their defects in perceiving basic table structures. We speculate that the way the visual encoder of MLLMs tokenizes input images could disrupt the integrity of table structures, e.g., traditional ViT models segment input images into fixed-size grid patch tokens, thereby resulting in fragmented table images. Developing advanced semantic equivalent visual tokenizers that can maintain the completeness and the structure of input table images may alleviate this problem [51].

(3) Combining with the experimental analysis of LLMs, we can infer an important fact: **although seemingly powerful large models including table-oriented models have achieved good per-**

Table 2: Performance of representative MLLMs on NIAT benchmark. 'Flat', 'Hori.' and 'Hie.' denote flat, horizontal and hierarchical tables, respectively. The best and suboptimal results are highlighted in **bold** and underlined.

| Model | Cell-Locating | | | | Cell-Lookup | | | |
|---|---|---|---|---|---|---|---|---|
| | Flat | Hori. | Hie. | Ave. | Flat | Hori. | Hie. | Ave. |
| GPT-4o | 28.46 | 15.34 | 11.4 | 18.40 | 66.9 | 61.43 | 76.65 | 68.33 |
| DeepSeek-VL2-tiny-3B | 5.89 | 4.08 | 5.47 | 5.15 | 55.29 | 25.44 | 54.68 | 45.14 |
| GLM-4V-9B-Chat | 4.73 | 3.72 | 2.68 | 3.71 | 52.63 | 24.41 | 50.00 | 42.35 |
| InternVL-2.5-8B-Chat | 15.53 | 10.76 | 12.44 | 12.91 | 37.89 | 33.37 | 24.39 | 31.88 |
| Llava-1.5-7B | 0.69 | 0.56 | 0.93 | 0.73 | 2.71 | 4.63 | 1.50 | 2.95 |
| Table-Llava-7B | 7.65 | 4.90 | 11.62 | 8.06 | 9.58 | 5.59 | 2.68 | 5.95 |
| Llava-1.6-7B | 2.57 | 1.61 | 3.50 | 2.56 | 16.88 | 5.39 | 7.80 | 10.02 |
| MiniCPM-V-2.6-8B | 23.99 | 16.21 | **20.12** | 20.11 | 59.50 | 36.10 | 37.37 | 44.32 |
| Phi-3.5-Vision-Instruct-4.2B | 2.28 | 2.61 | 4.13 | 3.01 | 41.09 | 22.17 | 36.48 | 33.25 |
| Phi-4-multimodal-instruct | 4.06 | 4.49 | 6.47 | 5.01 | 57.86 | 31.22 | 26.87 | 38.65 |
| Qwen2-VL-7B-Instruct | 5.53 | 4.43 | 5.07 | 5.01 | 63.04 | 21.83 | 53.24 | 46.04 |
| Qwen2.5-VL-3B-Instruct | 8.01 | 5.45 | 7.77 | 7.08 | 59.71 | 38.02 | 65.96 | 54.56 |
| Qwen2.5-VL-7B-Instruct | 10.53 | 7.92 | 9.39 | 9.28 | 70.24 | 51.83 | 72.32 | 64.80 |
| Llama-3.2-11B-Vision-Instruct | 4.46 | 4.25 | 5.99 | 4.90 | 37.35 | 36.46 | 51.74 | 41.85 |
| Qwen2.5-VL-32B-Instruct | 23.30 | 21.41 | 14.60 | 19.77 | 74.82 | 44.01 | 84.77 | 67.87 |
| Qwen2.5-VL-72B-Instruct | 25.80 | 25.13 | 18.60 | 23.18 | 72.72 | 48.94 | **87.68** | 69.78 |
| Skywork-R1V3-38B | **36.07** | 26.11 | 11.33 | 24.50 | 56.87 | **65.86** | 69.04 | 63.92 |
| GLM-4.1V-Thinking-9B | 27.77 | **31.54** | 18.24 | **25.85** | **76.93** | 64.15 | 84.28 | **75.12** |

**formance on many tabular benchmarks involving complex tasks, their success may not be grounded on a robust and reliable long-context table understanding ability**, i.e., they indeed can solve certain complex problems in existing benchmarks, but they still can not reliably perceive and understand the information of each cell in lengthy input tables. This also shows that existing models may rely on dataset-specific correlations or shortcuts to obtain better benchmark results or there may be serious data leakage problems for some models.

# 4 Experiments on Downstream Tasks

## 4.1 Data Synthesis Method

Based on above findings, we further explore **whether enhancing NIAT capabilities can fundamentally improve models' overall table understanding ability**. If the long-context table understanding truly lays the foundation of more advanced high-level table-related capacities, its improvements should also benefit the performance on downstream tasks. To this end, we propose a strong2weak data synthesis method to create NIAT fine-tuning data. Specifically, we collect Markdown tables from the training split, and generate various NIAT queries and corresponding responses using GPT-4o as instruction-tuning data. For cell-locating task, we adopt the same prompt templates in the NIAT benchmark and automatically construct queries based on randomly selected table cells. For cell-lookup task, to avoid overlapping with lookup questions in the test data of downstream benchmarks, we synthesize queiries of 6 more difficult lookup tasks to improve data diversity, which are listed in the Appendix C. These tasks present greater challenges than the simplest lookup questions, requiring LLMs to develop a more thorough understanding of table structures and header information. For instance, cell-retrieval task requires LLMs to perform lookup operations within the whole input table and identify all locations of a given cell content.

Besides, previous tabular LLMs often directly transform existing tabular datasets into fine-tuning data with short answers as target responses [31, 52], which can not provide enough supervised information for LLMs and may also lead to overfitting towards dataset-specific shortcuts. Therefore, we meticulously craft demonstrations and utilize GPT-4o with in-context learning to synthesize detailed chain-of-thought (CoT) reasoning processes as target responses, which aims at instructing weak LLMs to progressively understand tabular data and complete NIAT tasks. In the end, 6K instruction-tuning data were constructed for cell-locating and cell-lookup tasks, respectively.

Table 3: Performance results on downstream tasks. The best results in different model categories are highlighted in **bold**, and + *NIAT* denotes models further fine-tuned on our synthetic data.

| Method | NIAT Task | | | Downstream Task | | | | |
|---|---|---|---|---|---|---|---|---|
| | Cell-Locating | Cell-Lookup | Ave. Acc | WTQ | TabFact | HiTab | TABMWP | Ave. Acc |
| *Open-source LLMs* | | | | | | | | |
| GLM4-9B-Chat | **6.74** | 49.17 | 27.96 | **45.60** | 43.50 | 25.90 | 47.51 | 40.63 |
| MiniCPM3-4B | 1.18 | **58.13** | **29.66** | 40.58 | **62.88** | 24.43 | **69.39** | **49.32** |
| InternLM2.5-7B-Chat | 0.96 | 35.26 | 18.11 | 34.76 | 33.00 | 18.78 | 54.83 | 35.34 |
| Yi-1.5-9B-Chat | 4.41 | 49.05 | 26.73 | 34.00 | 45.40 | **32.70** | 34.93 | 36.76 |
| *Tabular LLMs* | | | | | | | | |
| StructLLM | 1.82 | 42.72 | 22.27 | 31.08 | 29.45 | 15.74 | 39.37 | 28.91 |
| TableLLM | - | - | - | 35.86 | 31.47 | 17.96 | 28.91 | 28.55 |
| TableGPT2 | **8.84** | **73.87** | **41.36** | **60.01** | **61.17** | **36.04** | **56.19** | **53.35** |
| *Ours* | | | | | | | | |
| Qwen2.5-7B-Instruct | 9.46 | 47.60 | 28.53 | 52.90 | 70.00 | 30.50 | 54.42 | 51.96 |
| +NIAT | **10.84** | 58.41 | 34.63 | 60.28 | 61.28 | **62.28** | **72.39** | 64.06 |
| Llama3.1-8B-Instruct | 6.16 | 65.74 | 35.95 | 49.90 | 62.80 | 26.10 | 54.78 | 48.40 |
| +NIAT | 8.38 | **66.46** | **37.42** | **67.43** | **78.57** | 49.41 | 66.15 | **65.39** |
| *Reasoning LLMs and GPT-4o* | | | | | | | | |
| GPT-4o | 26.02 | 68.30 | 47.16 | 83.50 | 65.80 | 39.10 | **84.38** | 68.19 |
| QwQ-32B | 35.42 | 49.74 | 42.58 | 78.26 | **75.23** | 61.35 | 53.76 | 67.15 |
| DeepSeek-R1 | **65.91** | **80.99** | **73.45** | **84.21** | 66.44 | **66.86** | 64.74 | **70.56** |

## 4.2 Experimental Setup

**Baselines.** Apart from open-source LLMs, reasoning LLMs like DeepSeek-R1, and close-sourced GPT-4o, we mainly compare our method with tabular LLMs that are fine-tuned with table instruction tuning data, including TableLLM [53] with 80K synthesized training data of downsteam tabular tasks, TableGPT2 [33] with 86B tokens for continuous pre-training (CPT) and 2.36M in-house data for table instruction tuning, and StructLLM fine-tuned with 1.1M samples of structured knowledge grounding tasks. We use the released model weights for inference, and fine-tune Llama3.1-8B-Instruct and Qwen2.5-7B-Instruct with 12K synthesized data as our methods.

**Benchmarks.** We evaluate model performance on 4 downstream benchmarks. WTQ [19] and HiTab [40] are TQA benchmarks based on flat tables and hierarchical tables, which contain lookup questions and reasoning questions that require additional aggregation operations. TabFact [20] is a TFV benchmark that requires models to verify whether a textual hypothesis holds based on the given evidence in a table. TABMWP [54] is a table-based math word problem benchmark that evaluates mathematical reasoning ability over structured tabular data. Tables are represented in the Markdown format for WTQ and TabFact, and HTML format for HiTab and TABMWP to maintain hierarchical table structures. We use exact match accuracy as the evaluation metric and report model performance under the zero-shot setting. **Complete implementation details are provided in Appendix C.**

## 4.3 Results and Analysis

Table 3 compares model performance on NIAT tasks and downstream tabular tasks. We can find that there is a substantial performance gap between NIAT tasks and more complex downstream tasks, demonstrating their weaknesses in the robust long-context table understanding ability. The proposed NIAT benchmark helps mitigate the data leakage issue among rapidly evolving LLMs, where training data and even test data in downstream tabular benchmarks could be excessively utilized for fine-tuning to achieve better benchmark results. We believe that a model with truly robust table understanding ability should not only perform well on traditional downstream tasks, but also maintain strong capability in simple but foundamental NIAT tasks, which is the foundation for reliable LLM-based table applications, especially in scenarios involving lengthy tables.

Compared to vanilla models, fine-tuning with our synthesized 12K NIAT data brings substantial performance enhancements for Qwen2.5-7B-Instruct and Llama3.1-8B-Instruct, improving the average

accuracy in both NIAT tasks and downstream tasks by 3.78% and 14.55%. Notably, our synthesized data does not contain training data of downstream benchmarks and solely focuses on enhancing the basic long-context understanding ability towards table structures and content. However, our fine-tuned models achieve better overall performance than baseline LLMs of similar scale, even surpassing tabular LLMs like TableGPT2 that use much more training data. This validates the effectiveness and efficiency of our synthesized data, and verifies our initial assumption that NIAT capabilities constitute the very foundation of more advanced table-related tasks. Figure 4 also intuitively demonstrates the improvement of our synthesized data on every cell position within cropped tables. **Due to space limitation, more experimental results such as ablation experiment are provided in Appendix D.**

### 4.4 Ablation Study

Table 4: Ablation results of different tasks further fine-tuned on *Llama3.1-8B-Instruct*. The best results are highlighted in **bold**. WTQ, TabFact, and HiTab are the three downstream datasets selected for fine-tuning with chain-of-thought (CoT) reasoning processes generated by GPT-4o on foundation LLMs, as listed in the table.

| Model | NIAT | | | | | | | WTQ | TabFact | HiTab | TABMWP |
|---|---|---|---|---|---|---|---|---|---|---|---|
| | 8 | 12 | 16 | 20 | 24 | 28 | 32 | | | | |
| *Llama3.1-8B-Instruct* | 16.88 | **13.56** | **12.94** | 8.98 | 6.34 | 7.25 | 3.83 | 49.90 | 62.80 | 26.10 | 54.78 |
| + Cell-Locating & Cell-Lookup | **20.10** | 12.75 | 10.21 | 5.51 | 4.51 | 4.88 | 2.49 | **67.43** | **78.57** | **49.41** | 66.15 |
| + Cell-Locating | 19.76 | 13.49 | 12.06 | 7.86 | 5.31 | 6.22 | 3.07 | 67.33 | 67.45 | 33.44 | 70.50 |
| + Cell-Lookup | 18.85 | 13.13 | 12.15 | 8.36 | 6.22 | 5.83 | 3.21 | 59.00 | 53.50 | 35.00 | 69.44 |
| + 4 downstream datasets | 16.28 | 11.12 | 11.41 | 6.11 | 5.92 | 5.40 | 2.17 | 64.78 | 78.13 | 48.22 | **81.79** |

We conduct ablation experiments to investigate the impact of different training tasks systematically. The experimental results are presented in Table 4. We compare the foundation model fully trained on our proposed mixed synthetic data to variants trained on individual sub-tasks, as well as to models fine-tuned on the training splits of WTQ, TabFact, and HiTab, which serve as oracle experiments representing the performance ceiling.

**Fine-tuning on our synthesized training data achieves comparable performance to variants fine-tuned on three downstream datasets.** Surprisingly, when fine-tuned on our mixed synthetic data, *Llama3.1-8B-Instruct* achieves comparable accuracy on the TabFact dataset and even outperforms other models on the remaining downstream tasks. For variants trained solely on Cell-Locating and Cell-Lookup tasks (+ Cell-Locating and + Cell-Lookup), removing NIAT training data significantly drops performance across all four downstream tasks.

## 5 Conclusion

This paper makes the first systematic exploration of long-context table understanding problem that has been overlooked by prior work, together with a new benchmark NEEDLEINATABLE, which covers tables of diverse structures, formats and sizes, and can serve as a valuable testbed for evaluating models' underlying fine-grained understanding and perception of individual cells within tabular context. On this basis, we conduct extensive evaluation of existing LLMs and MLLMs, revealing that current large models lack truly robust long-context table understanding ability, which could influence the reliability of LLM-based table applications. Furthermore, we propose a data synthesis method for enhancing LLMs' basic NIAT capabilities, and demonstrate its improvements in models' overall table understanding abilities, outperforming strong baselines including recent tabular LLMs.

## 6 Acknowldagment

This work was supported by the National Natural Science Foundation of China (Nos. 62472419, 62472420)

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

# A  Broader Discussion

## A.1  Limitations

Although this work presents a comprehensive exploration of the long-context table understanding problem, certain limitations could be alleviated by future research. First, the proposed benchmark focuses on a single table in English. The long-context understanding in multi-table scenarios with broader language coverage deserves future explorations. Second, real-world applications such as document understanding could also require processing the hybrid content of tabular, textual, and image data. Therefore, it is valuable to explore the long-context understanding ability towards data consisting of multiple modalities. Third, we build the NIAT benchmark based on tables from public datasets to save cost, which primarily contain medium-sized tables. Future work could collect larger tables that span diverse structures to provide a longer context for long-context table understanding, especially tables with more than thousands of rows.

## A.2  Ethical Considerations

The proposed NIAT benchmark is constructed based on the academic datasets like WTQ and TabFact, which are free and open datasets for research use with licenses like MIT License[5] or CC-BY-SA4.0 License[6]. The resulting synthetic training data is also a free and open resource for the community to study long-structured table understanding. Considering that LLMs may generate harmful content, we used Llama3.1-70B-instruct to conduct LLM-as-a-judge to check the content of synthetic data and the generated data by API-available models, and we also randomly sampled a part of the data for manual checking and did not observe unsafe data in the synthetic samples. Thus, the authors foresee no ethical concerns with the research in this paper.

## A.3  Different fine-tuning benefits between Qwen2.5 and Llama3.1

One notable finding from our experiments is the significant disparity in performance gains on the cell-locating task after NIAT fine-tuning (Figure 4). Specifically, Qwen2.5-7B demonstrated substantially greater improvement than Llama3.1-8B. We hypothesize this phenomenon is attributable to the models' differing pre-existing capabilities, stemming from two potential factors. First, the pre-training regimen of the models plays a crucial role. As detailed in its technical report [49], the Qwen2.5 series underwent specialized post-training on structured tabular data. This inherent familiarity with table structures may provide a more effective foundation for our CoT-based fine-tuning. Consequently, the fine-tuning process could more readily "activate" and enhance this latent ability, leading to a more pronounced boost in table structure comprehension. Second, existing research in reinforcement learning [55, 56] has indicated that the Qwen2.5 series possesses strong reasoning capabilities, even without instruction tuning. A superior innate reasoning capacity could make the model more adept at acquiring complex, multi-step procedural tasks, such as the cell-locating process of interpreting a table row-by-row and then pinpointing a cell by its column index. In contrast, the Llama3.1 series might require additional mid-training to achieve a comparable level of performance on such tasks.

## A.4

# B  Details of NIAT Benchmark

## B.1  Data Statistics

To construct a benchmark for long-context understanding of tabular data at a low cost, we leveraged existing open-source datasets to sample tabular data. Table 5 represents the statistics results of sampled tables, we calculate the average length of tokenized tabular data. Figure 5 shows the distribution of input length of the NIAT benchmark.

---

[5]https://opensource.org/license/mit/
[6]https://creativecommons.org/licenses/by-sa/4.0/deed.en

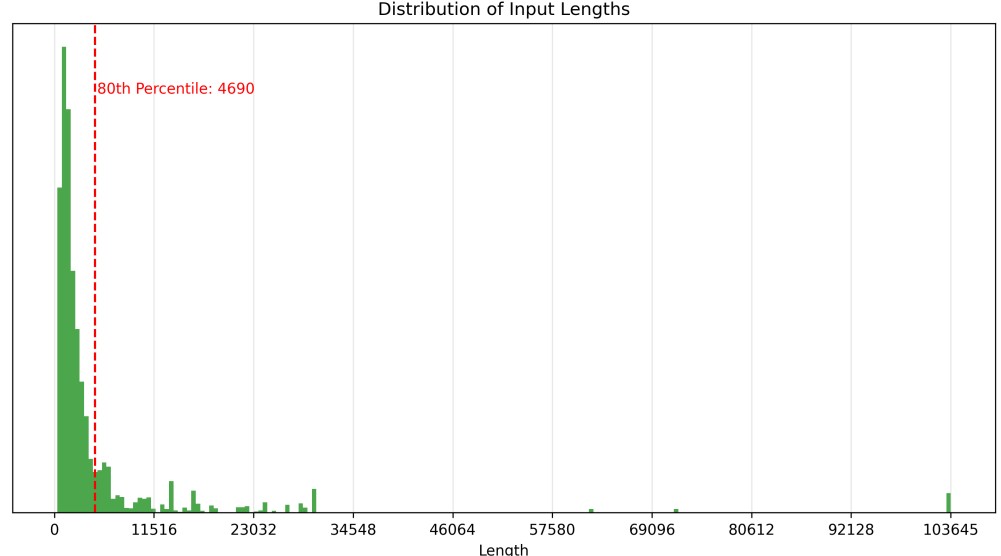

Figure 5: The distribution of input lengths for our proposed NIAT benchmark. The tokenizer of Llama3.1-8B-Instruct is adopted to calculate the token length.

Although the largest tables reach a length of 100k tokens, the average length of tables sampled from public dataset is not long enough. We anticipate that future researchers will build upon this work to develop more comprehensive benchmarks for long-table understanding tasks with longer tabular text.

| Table Types | Format | Table Num. | Avg. Row Num. | Avg. Col Num. | Avg. Length | Max Lengh |
|---|---|---|---|---|---|---|
| Horizontal | HTML | 325 | 6.43 | 26.78 | 1831.27 | 29791 |
| | Mardown | 358 | 6.38 | 25.77 | 981.92 | 16012 |
| Flat | HTML | 362 | 26.32 | 6.51 | 5226.04 | 103430 |
| | Markdown | 362 | 26.24 | 6.38 | 843.05 | 13203 |
| Hierarchical | HTML | 1086 | 19.81 | 8.87 | 878.39 | 5339 |
| | Markdown | 1086 | 19.81 | 8.87 | 1481 | 9420 |
| Total | - | - | 18.56 | 11.46 | 1594.32 | 103430 |

Table 5: Detailed table statistics of sampled tables of NIAT benchmark. The **Avg. Length** and **Max Length** represent the average and maximum token lengths of encoded tabular text for specific table categories. We use the tokenizer of Llama3.1-8B-Instruct to calculate the token length.

## B.2 Prompt Templates of Evaluating LLMs

For text-LLMs and MLLMs, we adopt unified prompt templates to evaluate the performance of them. For Cell-Locating the prompt templates are shown in Figure 9, for Cell-Lookup task, the adopted template are provided in Figure 10.

## C Implementation Details

### C.1 Implementation Details

To investigate the relationship between the fundamental table structure comprehension capabilities of LLMs and their performance in downstream table understanding tasks, we selected four popular datasets spanning various table understanding tasks. These datasets include **Table Question Answering** (HiTab, WTQ ), **Tabular Numerical Reasoning** (TABMWP), and **Table Fact Verification** (TabFact). Each language model was instructed to generate a chain-of-thought reasoning process before providing its final answer. To evaluate performance, we calculated accuracy based on the last 30 characters of the generated content for open-source LLMs, which corresponds to the model's final answer.

We provide an example of a Cell-locating question and answer pair, along with the chain-of-thought (CoT) reasoning process, in Figure 11.

## C.2 Training Details

To efficiently fine-tune *Qwen2.5-7B-Instruct* and *Llama3.1-8B-Instruct* on our synthetic data, we employ the NVIDIA Megatron framework. To avoid out-of-memory errors caused by long-context input sequences, we configure tensor parallelism (tp) to 8 and model pipeline (pp) to 4. Each model is fully fine-tuned on the mixed data for 2 epochs.

## C.3 Evaluation Details

For the Data Mixing Strategy, we randomly sample 8,400 NIAT training instances and 6,497 table-lookup question-answer pairs, combining them into the training dataset. The longest sample in our training data reaches 92k tokens. **For evaluating all foundation models and the further fine-tuned models**, we adopt the original generation configurations of the models and utilize the vLLM framework for deployment on a machine equipped with 8 NVIDIA A100-80G GPUs, as the input sequences can be extremely long.

## C.4 Details of Cell-Lookup Synthesis Tasks

**Locate single cell**    Given a row header and a column header, LLMs are tasked with identifying the cell value at the intersection of these two headers. The uniqueness of the ground truth answer is strictly enforced.
**Synthetic Data Generation Prompt Template**: f"In the table above, what is the element located in the cell at the intersection of the row header «row header» and the column header «column header»?"

**Retrieve the Nth cell based on row header**    Counting ability is crucial for LLMs when interpreting tabular data. In this task, given a row header (a specific cell value in the first column), the LLM is required to return the value of the Nth cell in the corresponding row.
**Synthetic Data Generation Prompt Template**: f"What is the Nth cell value with header row «row header»?"

**Retrieve the Nth cell based on column header**    : Similarly, LLMs are expected to locate the Nth cell value within a column specified by a given column header. Both tasks require LLMs to retrieve target cell contents simultaneously based on natural language queries and row/column IDs.
**Synthetic Data Generation Prompt Template**: f"What is the Nth cell value with header col «col header»?"

**Retrieve all location IDs of a certain key**    In some cases, cell values may be duplicated in a large table. This task requires identifying the row and column IDs (location IDs) of all cells that contain the target value.
**Synthetic Data Generation Prompt Template**: f"In the table above, how many cells contain the value «cell to ask»?"

**Cell value counting**    In this task, LLMs are asked to count the number of cells that contain a specific value, further testing their ability to process and summarize tabular data.
**Synthetic Data Generation Prompt Template**: f'How many cell contains the value «random element»? Please provide all row IDs and column IDs of all cells contain "«random element»".'

**Table navigation**    Understanding the relative positions of two different cells in a table is a challenging task for LLMs. Given a specific base cell position and row/column offsets, LLMs are required to navigate the two-dimensional table by calculating the target position IDs and retrieving the corresponding cell values.
**Synthetic Data Generation Prompt Template**:"You are provided with a two-dimensional table and need to locate the content of a specific cell. The following information is given: Base Position: Row index: «base row» Column index: «base col» Relative Position: Row offset: «relative row» Column offset: «relative col» Search Instructions: If «relative row» > 0, move downwards from the base row index. If «relative row» < 0, move upwards from the base row index. If «relative col» > 0, move rightwards from the base column index. If «relative col» < 0, move leftwards from the base column index. "Calculate the new target position («new row», «new col») using these offsets from the base position, and return the content of the cell located at this new position"

| | Format | Avg. Queries Num. |
|---|---|---|
| Horizontal | HTML | 317.56 |
| | Markdown | 376.33 |
| Flat | HTML | 290.57 |
| | Markdown | 372.67 |
| Hierarchical | HTML | 909 |
| | Markdown | 1041 |

Table 6: The average number of queries of question-based NIAT. We prompt GPT-4o to generate simple questions to extract cells of a given table and check the correctness of

**Prompt Template of 6 Synthesized Cell-Lookup tasks**    We randomly sampled to examples of Cell-Lookup training data synthesized by GPT-4o are shown in Figure 12 (Retrieve the Nth cell based on column header) and 13 (Table navigation).

# D   Additional Experiments

## D.1   Results on Cropped Tables

Table 8 presents the detailed performance metrics of various MLLMs. In Table 7, we compare LLMs fine-tuned on our proposed NIAT synthetic data against mainstream LLMs and TableLLMs on Cell-Locating tasks with cropped tables. This task demands a deep structural understanding of two-dimensional tabular data, so we evaluated model performance across varying context window sizes. Consistent with expectations, longer context windows improve accuracy in Cell-Locating tasks.

The **Qwen2.5 series**dominates performance, likely due to its specialized training on tabular data (as noted in its technical report). Notably, **Qwen2.5-7B-Instruct-1M**, a long-context variant, significantly outperforms its standard counterpart (**Qwen2.5-7B-Instruct**), suggesting benefits from structured pre-training and post-training data.

Among **Thinker LLMs**, DeepSeek-R1 achieves state-of-the-art results, leveraging its Long-CoT capability to accurately retrieve target cells in large 32×32 tables using row/column identifiers. This underscores the potential of test-time scaling techniques for processing long-structured tabular data. Surprisingly, Qwen2.5-7B-Coder-Instruct (with only 7B parameters) excels on NIAT tasks, likely due to: 1) Structural alignment between tabular data and code (both rely on separators for semantic organization). 2) Extensive pre-training on markdown, which may have included markdown tables, priming it for Cell-Locating NIAT tasks.

**Multi-Modal LLMs (MLLMs)**, which process table images directly, more closely mirroring human tabular understanding, demonstrate superior performance in Cell-Locating tasks. Notably, recent state-of-the-art models like Qwen2.5-VL-7B-Instruct and MiniCPM-V-2.6.8B significantly outperform earlier approaches (e.g., Llava-1.5-7B), while they still underperform GPT-4o with pure text input.

As illustrated in Figure 6, both mainstream **MLLMs** and **text-LLMs** exhibit the **lost-in-the-middle-tables** phenomenon—their performance degrades monotonically as table size increases. For MLLMs, this limitation arises because visual encoders struggle to accurately parse table structures when rows and columns expand beyond 32×32, leading to unreliable target cell retrieval. Nevertheless, recent MLLMs (e.g., MiniCPM-V-2.6-8B) show marked improvements in Cell-Locating tasks, suggesting advances in structural understanding.

## D.2   Case Study

We provide two cases of the model further fine-tuned on our synthesized data on **Cell-Locating** and **Cell-Lookup**, respectively in Figure 7 and 8.

Table 7: Performance results on Cropped Tables of mainstream text-LLMs, TableLLM, MLLMs, and foundation models fine-tuned on our synthetic data. The best results are highlighted in **bold**, and + *ours* denotes models further fine-tuned on our proposed synthetic data.

| Model | Cell-Locating on Cropped Tables | | | | | | |
|---|---|---|---|---|---|---|---|
| | **8** | **12** | **16** | **20** | **24** | **28** | **32** |
| *Open Source LLMs* | | | | | | | |
| GLM4-9b-chat | 9.9 | 10.88 | 10.26 | 8.43 | 4.91 | 8.65 | 4.29 |
| GLM4-9b-chat-1M | 5.63 | 6.39 | 5.94 | 5.43 | 3.84 | 3.99 | 2.57 |
| MiniCPM3-4B | 1.98 | 4.35 | 5.73 | 5.75 | 4.28 | 9.51 | 5.02 |
| InternLM2.5-7B | 4.9 | 6.25 | 4.84 | 5.23 | 3.24 | 3.85 | 2.25 |
| Mistral-7B-v0.3 | 8.54 | 10.65 | 9.17 | 6.2 | 3.47 | 5.49 | 5.81 |
| Yi-1.5-9B-16K | 12.81 | 8.98 | 5.55 | 5.67 | 3.32 | 4.53 | 2.12 |
| *Qwen2.5-Instruct* | | | | | | | |
| Qwen2.5-7B-Instruct | 26.35 | 18.98 | 15.91 | 12.83 | 7.55 | 9.72 | 2.9 |
| Qwen2.5-7B-Instruct-1M | 32.08 | 24.44 | 20.6 | 13.8 | 9.99 | 12.13 | 4.93 |
| Qwen2.5-14B-Instruct | 28.96 | 19.81 | 19.69 | 15.93 | 14.38 | 17.54 | 9.31 |
| Qwen2.5-14B-Instruct-1M | **39.69** | **33.84** | **33.83** | **26.7** | **20.98** | **25.54** | 16.54 |
| Qwen2.5-7B-Instruct + ours | 37.60 | 30.83 | 25.76 | 17.67 | 18.66 | 18.24 | 10.85 |
| *Llama3.1-Instruct* | | | | | | | |
| Llama-3.1-8B-Instruct | 16.88 | **13.56** | **12.94** | **8.98** | **6.34** | **7.25** | **3.83** |
| Llama-3.1-8B-Instruct + ours | **20.1** | 12.75 | 10.21 | 5.51 | 4.51 | 4.88 | 2.49 |
| *TableLLM* | | | | | | | |
| TableLLM | - | - | - | - | - | - | - |
| StructLLM | 5.62 | 5.23 | 4.29 | 4.02 | 2.07 | 3.26 | 0.92 |
| TableGPT2 | 21.15 | 15.46 | 12.99 | 10.12 | 8.34 | 9.57 | 3.4 |
| *Thinker LLM and Coder LLM* | | | | | | | |
| GPT-4o | 50.21 | 33.36 | 30.89 | 24.15 | 27.54 | 26.01 | 17.32 |
| QwQ-32B | 51.15 | 45.23 | 43.91 | 14.57 | 6.01 | 6.32 | 6.56 |
| DeepSeek-R1 | **73.33** | **64.63** | **64.60** | **41.87** | **38.07** | **54.06** | **51.17** |
| Qwen2.5-7B-Coder | 65.42 | 54.86 | 44.90 | 28.42 | 17.79 | 29.88 | 28.67 |
| *MLLMs* | | | | | | | |
| GPT-4o(Text) | 41.67 | 18.34 | 19.94 | 9.75 | 8.84 | 18.46 | 3.92 |
| GLM-4V-9B | 2.80 | 2.61 | 4.30 | 1.39 | 1.43 | 0.89 | 0.09 |
| InternVL-2.5-8B | 28.52 | 16.64 | 12.60 | 10.24 | 10.96 | 13.18 | 5.09 |
| Llava-1.5-7B | 2.68 | 4.31 | 1.19 | 0.20 | 1.26 | 0.39 | 0.30 |
| Table-Llava-7B | 1.67 | 2.37 | 2.65 | 2.12 | 2.97 | 1.82 | 2.01 |
| Llava-1.6-7B | 6.60 | 4.03 | 4.24 | 5.57 | 4.11 | 4.33 | 0.26 |
| MiniCPM-V-2.6-8B | 43.51 | 30.57 | 15.57 | 11.74 | 10.01 | 12.88 | 3.94 |
| Phi-3.5-Vision-Instruct-4.2B | 4.25 | 2.84 | 5.92 | 2.68 | 4.68 | 1.56 | 0.82 |
| Phi-4-multimodal-instruct | 6.04 | 8.86 | 8.84 | 5.77 | 6.12 | 8.69 | 3.68 |
| Qwen2-VL-7B-Instruct | 7.27 | 8.72 | 8.62 | 9.64 | 8.19 | 8.07 | 2.30 |
| Qwen2.5-VL-3B-Instruct | 15.66 | 13.60 | 10.95 | 10.53 | 9.70 | 8.11 | 2.67 |
| Qwen2.5-VL-7B-Instruct | 19.80 | 14.64 | 11.98 | 10.67 | 5.29 | 9.59 | 3.02 |
| Llama-3.2-11B-Vision-Instruct | 10.63 | 8.96 | 9.89 | 6.98 | 4.34 | 5.30 | 3.03 |

Table 8: Performance results of mainstream MLLMs on cropped tables. The best results are highlighted in **bold**.

| Model | Cell-Locating NIAT task on cropped tables | | | | | | | |
|---|---|---|---|---|---|---|---|---|
| | **8*8** | **12*12** | **16*16** | **20*20** | **24*24** | **28*28** | **32*32** | **Ave.** |
| *Multi-Modal LLMs* | | | | | | | | |
| *DeepSeek-VL2-tiny-3B* | 2.8 | 4.08 | 2.46 | 4.27 | 3.47 | 2.49 | 1 | 2.94 |
| *GLM-4V-9B* | 2.8 | 2.61 | 4.3 | 1.39 | 1.43 | 0.89 | 0.09 | 1.93 |
| *Llava-1.5-7B* | 2.68 | 4.31 | 1.19 | 0.2 | 1.26 | 0.39 | 0.3 | 1.48 |
| *Table-Llava-7B* | 1.67 | 2.37 | 2.65 | 2.12 | 2.97 | 1.82 | 2.01 | 2.23 |
| *Llava-1.6-7B* | 6.6 | 4.03 | 4.24 | 5.57 | 4.11 | 4.33 | 0.26 | 4.16 |
| *Phi-3.5-Vision-Instruct-4.2B* | 4.25 | 2.84 | 5.92 | 2.68 | 4.68 | 1.56 | 0.82 | 3.25 |
| *Phi-4-multimodal-instruct-5.6B* | 6.04 | 8.86 | 8.84 | 5.77 | 6.12 | 8.69 | 3.68 | 6.86 |
| *Qwen2-VL-7B-Instruct* | 7.27 | 8.72 | 8.62 | 9.64 | 8.19 | 8.07 | 2.3 | 7.54 |
| *Qwen2.5-VL-3B-Instruct* | 15.66 | 13.6 | 10.95 | 10.53 | 9.7 | 8.11 | 2.67 | 10.17 |
| *Qwen2.5-VL-7B-Instruct* | 19.8 | 14.64 | 11.98 | 10.67 | 5.29 | 9.59 | 3.02 | 10.71 |
| *Llama-3.2-11B-Vision-Instruct* | 10.63 | 8.96 | 9.89 | 6.98 | 4.34 | 5.3 | 3.03 | 7.02 |
| *MiniCPM-V-2.6-8B* | **43.51** | **30.57** | 15.57 | **11.74** | 10.01 | 12.88 | 3.94 | **18.32** |
| *InternVL-2.5-8B* | 28.52 | 16.64 | 12.6 | 10.24 | 10.96 | 13.18 | **5.09** | 13.89 |
| *GPT-4o* | 41.67 | 18.34 | **19.94** | 9.75 | 8.84 | **18.46** | 3.92 | 17.27 |

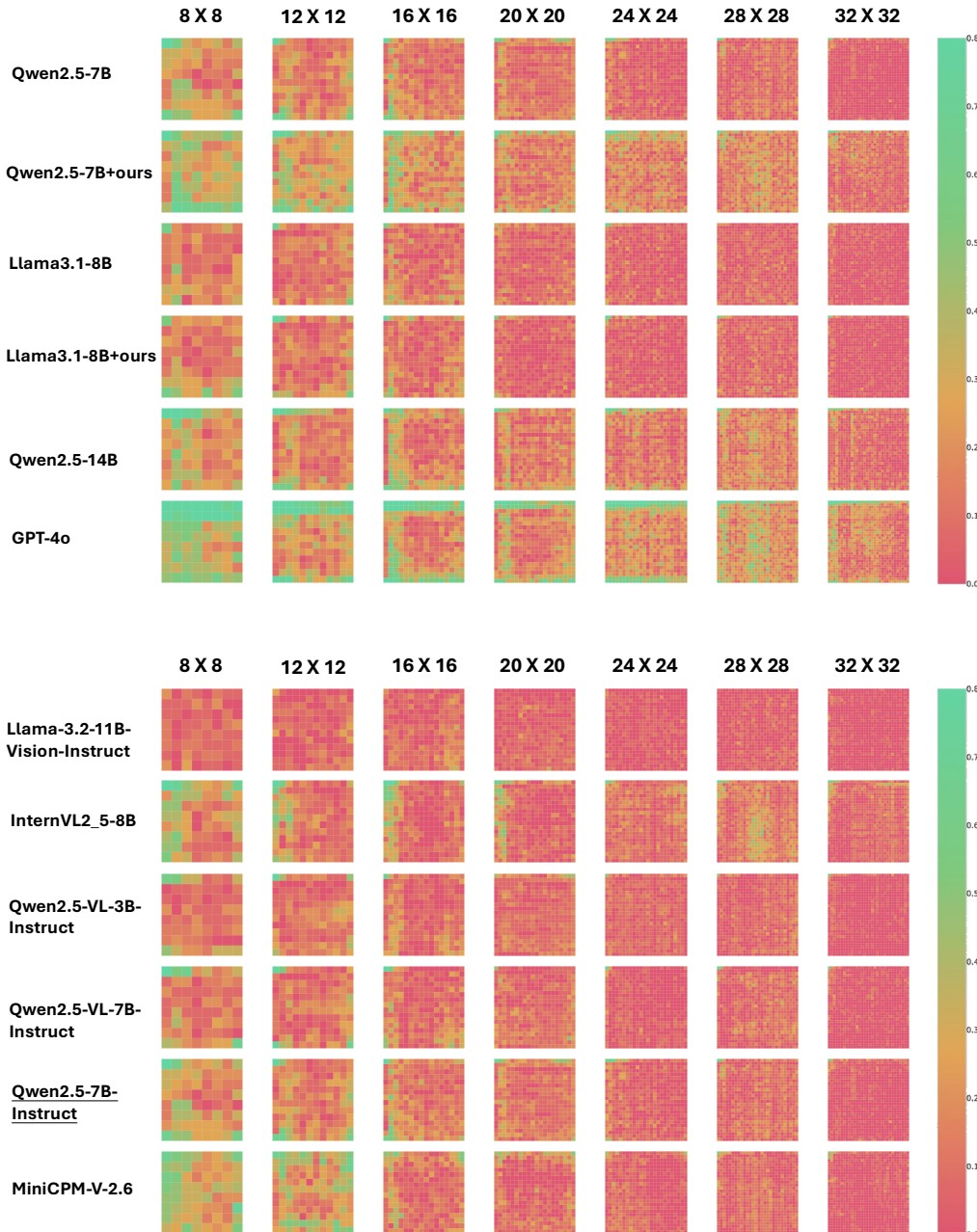

Figure 6: The figure presents a NEEDLEINATABLE evaluation performance comparison on cropped tables between MLLMs and text-LLMs.

```
An example of finetuned Llama3.1-8B-Instruct output:
To determine the value at the specified cell, we will follow these steps:

1. Identify the Row:
   - We are looking for the row with a row ID of 8. In the provided table
   , the rows are numbered from 1 to 8, corresponding to each province
   listed.
   - The row with a row ID of 8 corresponds to the province "Seville".

2. Identify the Column:
   - We are looking for the column with a column ID of 1. The columns are
    labeled as follows:
     - Column 1: Province
     - Column 2: Capital
     - Column 3: Population
     - Column 4: Density
     - Column 5: Municipalities
     - Column 6: Legal districts

3. Locate the Cell:
   - With the identified row (Seville, row ID 8) and column (Province,
   column ID 1), we locate the cell at the intersection of these two.
   - The cell at row ID 8, column ID 1 contains the value "Seville".

4. Output the Result:
   - The cell value located at the specified position is "Seville".

Therefore, the final answer is:

```
{"row_ID": 8, "column_ID": 1, "cell_value": "Seville"}
```

This matches the provided ground truth answer, confirming that the value
in the specified cell is "Seville".
```

Figure 7: A case response from Llama3.1-8B-Instruct, further fine-tuned on synthesized data, demonstrating its performance on the Cell-Locating task.

```
An example of finetuned Llama3.1-8B-Instruct output:
To determine the opponent in Week 16, we need to carefully examine the
table provided. Let's go through a step-by-step reasoning process:

1. Identify Week 16:
   - We need to locate Week 16 in the table. The weeks are listed
   sequentially from Week 1 to Week 17. We look for the entry that
   corresponds to Week 16.

2. Locate the Date for Week 16:
   - In the table, the date for Week 16 is December 23, 2007.

3. Find the Opponent for Week 16:
   - Once we have identified the correct week, we look at the "Opponent"
   column for that row. For Week 16, the opponent listed is "Philadelphia
    Eagles."

4. Verify the Details:
   - To ensure accuracy, we check the other details in the same row to
   confirm consistency. The "Game Site" is "Louisiana Superdome," the "TV
   " channel is "Fox," and the "Result/Score" is "L 38-23," which matches
    the pattern of losses (L) followed by a score in the format of "Score
    - Score."

5. Conclusion:
   - Based on the information extracted from the table, the opponent in
   Week 16 is confirmed to be the Philadelphia Eagles.

Final Answer:
```json
{{
    "answer": "Philadelphia Eagles"
}}
```
```

Figure 8: A case response from Llama3.1-8B-Instruct, further fine-tuned on synthesized data, demonstrating its performance on the Cell-Lookup task.

```
Read the table image carefully and retrieve the value from a specified
cell, based on a given row ID and a column ID.
Let's first think step by step and then output a final answer in the
following JSON format:

```json
{{
    "row_ID": "m",
    "column_ID": "n",
    "cell_value": "<Corresponding Cell Value>"
}}
```

Please note:
-  Row and column IDs start from 1.
-  The header row is considered as the first row, with a row_ID of 1, so
ensure you include it when locating the target row.
-  If the target cell is empty, the cell_value should be an empty string
("").

Now, retrieve the cell value at the position where the row_ID is {} and
the column_ID is {}.

## Your Response:
```

Figure 9: Prompt template for evaluating Cell-Locating Task.

```
Read the table image carefully and answer the following question.
Let's first think step by step and then output a final answer in the JSON
 format:
```json
{{
    "answer": "<the final answer to the question>"
}}
```
## Question:
{}

## Your Response:
```

Figure 10: Prompt template for evaluating Cell-Locating Task.

32

Figure 11: An example of the chain-of-thought (CoT) reasoning process for retrieving a target cell using row ID and column ID, step by step.

```
Carefully review the table, question, and provided ground truth answer.

Table: <table_str>

Question: In the table above, what is the element located in the cell at
the intersection of the row header 12 and the column header game?

Ground truth Answer: 12
Generate a detailed, step-by-step reasoning process to accurately answer
the question. Conclude with the final answer.

GPT-4o Answer:
To determine the element located in the specified cell, follow these
steps:

1. Understand the Table Structure:
   - The table is organized with columns labeled as: `game`, `date`, `
   team`, `score`, `high points`, `high rebounds`, `high assists`, `
   location attendance`, and `record`.
   - Each row corresponds to a particular game's data, with the rows
   numbered from 1 to 13.

2. Identify the Cell to Examine:
   - The question asks for the element at the intersection of the row
   header 12 and the column header `game`.
   - The row header refers to the game's sequence or identifier within
   this particular dataset; the row number is essentially the order of
   games played.

3. Locate Row 12:
   - Go to the row labeled with the number 12 in the table. This
   corresponds to all the data associated with the 12th game played.

4. Find Column Header `game`:
   - The `game` column is the first column in the table. This column
   indicates the game number or identifier for that particular set of
   data.

5. Extract the Information:
   - At the intersection of row 12 and the column `game`, locate the
   number, which indicates the sequence or position of that game.

6. Interpret the Table Data:
   - If we review the table, we can see that in row 12 under the column `
   game`, the cell contains the number "12". This means that the 12th row
    corresponds to the 12th game played.

Finally, using this step-by-step examination, the element located in the
cell at the intersection of the row header 12 and the column header `game
` is indeed:

Final Answer: 12
```

Figure 12: An example of the chain-of-thought (CoT) reasoning process for synthesized Cell-lookup tasks.

```
Carefully review the table, question, and provided ground truth answer.

Table: <table_str>

Question:
You are provided with a two-dimensional table and need to locate the
content of a specific cell. The following information is given:

Base Position:

Row index: 32
Column index: 4
Relative Position:

Row offset: -17
Column offset: 0
Search Instructions:

If relative_row > 0, move downwards from the base row index.
If relative_row < 0, move upwards from the base row index.
If relative_col > 0, move rightwards from the base column index.
If relative_col < 0, move leftwards from the base column index.
Calculate the new target position (new_row, new_col) using these offsets
from the base position, and return the content of the cell located at
this new position

Ground truth Answer: One of the original 19 parishes.
Generate a detailed, step-by-step reasoning process to accurately answer
the question. Conclude with the final answer.

GPT-4o Answer:
To determine the content of a specific cell using the given information,
we need to follow the search instructions carefully. Let's go through
each step:

1. Identify the Base Position:
   - The base row index is 32.
   - The base column index is 4.

2. Understand the Relative Position:
   - The row offset is -17, meaning we need to move 17 rows upwards from
   the base row.
   - The column offset is 0, meaning there is no movement in the columns,
    and we stay in the same column as the base column.

3. Apply the Row Offset:
   - Starting from the base row index of 32, move 17 rows upwards.
   - Calculate the new row index: \(32 - 17 = 15\).

4. Apply the Column Offset:
   - The base column index is 4 and the column offset is 0, so the new
   column index remains 4.

5. Locate the New Position in the Table:
   - The new row index is 15.
   - The new column index is 4.

6. Find the Content of the Cell at the New Position:
   - Check the table at row index 15 and column index 4.
   - According to the table, this corresponds to the "Origin" column for
   the 16th parish in the list, which is East Baton Rouge Parish since
   the table index starts from zero.

7. Verify and Extract the Content:
   - The content at row index 15 and column index 4 is "One of the
   original 19 parishes."

8. Conclusion:
   - The content of the cell located at the new position is "One of the
   original 19 parishes."

Thus, the final answer is:

"One of the original 19 parishes."
```

34

Figure 13: An example of the chain-of-thought (CoT) reasoning process for synthesized Cell-lookup tasks.