# OpenReview forum: "NeedleInATable: Exploring Long-Context Capability of Large Language Models towards Long-Structured Tables"
_NeurIPS.cc/2025/Conference — NeurIPS 2025 poster_

### Official Review · Reviewer_QGzx · 2025-06-10

**Clarity:** 3
**Significance:** 2
**Originality:** 3
**Rating:** 4
**Confidence:** 3

**Summary:**

The paper is concerned with evaluating LLMs and MLLMs on structured data.
A benchmark called NIAT (Needle in a Table) is proposed that tries to evaluate the fine-grained, cell-based understanding of these models.
This is achieved using two tasks cell lookup and cell locate, trying to find a needle (the cell) in a table.
Multiple LLMs and  MLLMs fail at this seemingly simple task while being able to do more difficult reasoning tasks on the tables.
A dataset (similar to the benchmark task) is constructed  and fine-tuning on this dataset improves performance.
Finally, the paper conducts analysis and ablation using the benchmark and dataset. Notably, it reports similar findings to lost in the middle but for tabular data.

**Questions:**

- The models under-perform in locate task compared to  look-up task. It seems like more of a "geometry" problem than table understanding problem (The model might understand the table but have difficulty finding 5th column 7th row). There are various other encodings (some of which would not even support cell based referencing). This raises two questions 1) Why is locating a cell important? 2) Would other encodings (like JSON, Data Matrix, DFLoader [1]) mitigate or solve the problem?
- In section 4.1 for the locate task the train/test split is used while for lookup, some other datasets are used. It is stated the reason is "to avoid overlapping with lookup questions in the test data of downstream benchmarks". What explains the difference between locate and lookup?
- Can more details be shared regarding Figure 3? What layer, model, size is used? Is it one example or average of many? Does the pattern occur across various models and layers and encodings?

[1] Fang, Xi, et al. "Large Language Models (LLMs) on Tabular Data: Prediction, Generation, and Understanding--A Survey." arXiv preprint arXiv:2402.17944 (2024).

**Ethical Concerns:**

["NO or VERY MINOR ethics concerns only"]

**Final Justification:**

After discussion some of my concerns were addressed and I decided to increase the rating from 3 to 4.

**Limitations:**

yes

**Paper Formatting Concerns:**

- Line 162 Needle in a Haystack should change to Needle in a Table.
- In the caption of Table 2, suboptimal should change to runner up or second best.

**Quality:**

2

**Strengths And Weaknesses:**

# Strengths
- The proposed method is general and applicable to LLMs, MLLMs and (some) models for tabular data.
- Identifies a simple failure mode (cell value by location).

# Weaknesses
- The paper tries to address many subjects while not going deep into any. (New benchmark creation that is applicable to wide range of models, Analysis on size and location, long context behavior, creating a new dataset and fine-tuning. details below)
- Long context behavior is not sufficiently explored. 80\% of the tables are less than 4k tokens. The largest table in lost in the middle analysis is 32 x 32. Assuming 10 token per cell, this results in ~10k tokens which is not long context.
- The part regarding attention pattern analysis (Figure 3) seems under-explored. It is not clear which model on which task and which layer is used. Is the pattern repeated in many examples?
- The models used for evaluation (both LLM and MLLM) seem limited in scope. Many of them have newer variant of the same size and larger variants with better performance. (The same 3k sub-sample method can be used to mitigate cost.)
- The claims in lines 257 \~ 259 and in 251 \~ 255 are not completely justified due to the above points. Various models like PaliGemma can perform segmentation even with image patches. To claim "their success may not be grounded on a robust and reliable long-context table understanding ability" requires investigating various table encodings, model sizes and more long context analysis.
- There are mismatches between main text and appendix. Main text reports 120k to be longest context while Table 1 in B.1 says otherwise. (different tokenizers?)
- The results of fine-tuning in Table 5 (ablation in appendix) are not conclusive and while there are some improvement in some tasks, some times there is decreased performance for NIAT itself (especially for smaller tables.)
- The paper claims success in some tasks might be due to leakage (line 263) yet it uses existing tables to create the benchmark.

---

> ### Author Rebuttal · Authors · 2025-07-31
>
> We sincerely thank you for the precious effort spent in carefully reviewing our paper. The questions will be properly addressed as we reply and your valuable advice will be fully incorporated.
>
> **Q1: The paper tries to address many subjects while not going deep into any.**
>
> **A1:** We totally understand the concern about the research depth in this work. However, **facing the gap in existing long-context and tabular benchmarks regarding long-context table understanding problem, we think that taking a solid first step is equally or even more important.** In this work, we not only introduce the new NIAT task and construct a large-scale benchmark for future study, but also conduct heavy and massive experiments to evaluate many LLMs and MLLMs with 300K+ samples, revealing their pros and cons in this new problem and providing valuable insights. Moreover, we also attempt to improve models' performance with a simple yet effective data synthesis method, serving as a fundamental baseline for following methods. **Therefore, we strongly believe that we make solid contributions towards this new problem with necessary in-depth investigations. Nevertheless, we also think this new problem deserves future efforts more than just one paper and we would like to further going deep into this problem**.
>
> **Q2: The NIAT table is not long enough to reach extremely large context-window.**
>
> **A2:** As mentioned in the Limitation section, to provide the community with the first benchmark on the long-context ability of structured tables with limited resources, we build NIAT benchmark based on tables from public datasets. The experimental results have shown that some powerful models struggle within these medium-sized tables even small tables, thus we anticipate that models would suffer more significant performance decline on longer and larger tables. We agree that more complex tasks together with larger tables and more hybrid content (like table-text scenario) should be considered to provide a more thorough evaluation of long-context table understanding, which we leave to future work.
>
> **Q3: The experimental settings about attention pattern analysis (Figure 3). Which model on which task and which layer is used? Is the pattern repeated in many examples?**
>
> **A3:** The input samples are from the WTQ benchmark (TQA task) and tables are serialized into Markdown format. We observe attention patterns from different attention heads from different layers of Llama3.1-series models, where we found that the 'Multi-Slash' attention pattern prevalently exist in the upper layers (20+ layers) while the 'Local-Triangle' attention pattern could exist in each layer. The illustrated attention patterns are not anomalies and are observed in many examples (100+). We will add these experimental details in the next version together with the corresponding attention distribution images.
>
> **Q4: Evaluating recent models of new version and larger variants.**
>
> **A4:** **Based on the sampled 6K test data, we evaluate some recent models especially those with R1-style reasoning abilities such as GLM-4.1V-Thinking and Qwen3 (in no-thinking mode).** From the results provided below, we can find that recent models achieve better performance on NIAT benchmark with enhanced reasoning abilities, but there is still room for improvements.
>
> | Models | Cell-Locating |  |  |  | Cell-Lookup |  |  |  |
> | --- | --- | --- | --- | --- | --- | --- | --- | --- |
> |  | Flat | Hie. | Hori. | Ave. | Flat | Hie. | Hori. | Ave. |
> | Qwen2.5-VL-32B-Instruct | 23.30 | 14.60 | 21.41 | 19.77 | 74.82 | 84.77 | 44.01 | 67.87 |
> | Qwen2.5-VL-72B-Instruct | 25.80 | **18.60** | 25.13 | 23.18 | 72.72 | **87.68** | 48.94 | 69.78 |
> | Skywork-R1V3-38B | **36.07** | 11.33 | 26.11 | 24.50 | 56.87 | 69.04 | **65.86** | 63.92 |
> | GLM-4.1V-Thinking-9B | 27.77 | 18.24 | **31.54** | **25.85** | **76.93** | 84.28 | 64.15 | **75.12** |
> |  |  |  |  |  |  |  |  |  |
> | Qwen3-14B | 18.22 | 2.61 | 22.00 | 13.99 | 72.60 | 76.20 | 63.00 | 70.60 |
> | Qwen3-32B | **22.51** | **3.52** | 27.67 | **17.34** | 73.90 | 80.31 | 66.50 | 73.57 |
> | Qwen3-30B-A3B | 21.82 | 2.51 | 25.47 | 16.48 | **78.90** | **85.39** | **71.02** | **78.44** |
>
> **Q5: The calculated benchmark prompt length differs between main body and appendix.**
>
> **A5:** In section 3.2 Benchmark Statistics in the main body, we reported an rounded word number of input prompt by whilespace-splited tokenizer (line 165). In appendix, the accurate token length of input prompt by Llama3.1-8B-Instruct tokenizer is reported, leading to a mismatch which will be unified in the next version.
>
> **Q6: In Table 5 (ablation study), fine-tuning with NIAT data brings improvements on some tasks, but not bring consistent improvements in cell-locating tasks on cropped tables of all sizes.**
>
> **A6:**  This is indeed a curious phenomenon and we present our results honestly. As shown in Figure 4, we can find that NIAT fine-tuning brings significant performance increase for Qwen2.5-7B but less gains and even slightly degeneration for Llama3.1-8B in cell-locating tasks on some cropped tables. For one thing, one potential reason is that the used cropped tables (15 tables for each size) for evaluation are still limited in amount, thus leading to normal performance fluctuation. From Table 3 in main body, we observe that fine-tuning with synthesized data effectively improves Llama3.1-8B performance in two NIAT tasks when evaluated with full 287K test data.
>
> For another, this could be attributed to models' pre-existing capabilities. On the one hand, Qwen2.5-series underwent specialized post-training on structured tabular data [1], which may lead to better table structure understanding ability that our CoT-based fine-tuning data could more effectively "activate" and build upon, leading to a significant boost in table structure comprehension. On the other hand, exisiting RL studies [2, 3] have shown that Qwen2.5-series possesses better reasoning ability, which makes it more easier to incentivize advanced capabilities for complex multi-step tasks, e.g., interpreting the table structure row by row and then pinpointing the answer cell by column index. By contrast, Llama3.1-series needs extra mid-training to achieve the comparable performance. We will perform ablation studies with Qwen2.5-series to further investigate this issue.
>
> **Q7: Why cell-locating is important? Can using other table formats like DataFrame mitigate or solve the problem?**
>
> **A7:** We fully understand the concern that cell-locating seems to be a trivial task compared to more complex tabular tasks like data analysis or table manipulation. However, **we think it serve as a fundamental probe task to evaluate models' basic understanding of diverse table structures, just like many efforts that have been spent to identified simple yet important failure modes of seemingly perfect (M)LLMs**. For instance, an MLLM is asked to analyze a table in a document image, how can we fully trust its results if it even does not know what is the content at the 1st row and 1st column.
>
> **Using other table formats (like DataFrame) with code-based solutions (like Text2SQL or Python) can indeed mitigate this problem in certain scenarios.** (1) For scenarios which only consider flat tables or database-like tables, we also think that semantic parsing methods like Text2SQL are preferable [4] as they can manipulate tables with executable code. (2) However, real-world application scenarios often involve processing large tables with complex structures, such as hierarchical headers[5], merged cells, nested tables and even multiple tables within a single document[6], whose table formats could not support standard processing of DataFrame-like semantic parsing approaches. Such practical use cases make it valuable and necessary to study the LLM-based understanding ability of long and structured tables of diverse formats within its context window.
>
> **Q8: Why synthesize cell-lookup questions of new types for NIAT fine-tuning?**
>
> **A8:**  Our synthesized NIAT fine-tuning data are based training tables from public datasets. As the downstream benchmarks like WTQ could also contain lookup questions, we generate cell-lookup questions of new types for NIAT fine-tuning to maximumly avoid the overlap between training data and test data in benchmarks.
>
> **Q9: Potential data contamination of used tables from previous datasets.**
>
> **A9:** The NIAT queries are constructed based on tables from previous academic datasets, which could indeed have been used in the pre-training stage of some proprietary models like GPT-4o and even some open-source models (as they did not fully open-source their training data). Nevertheless, none of these models achieved a near-perfect performance on the NIAT benchmark especially on the cell-locating task, which indicates that only memorizing table content during pre-training does not guarantee a robust fine-grained table understanding ability. In the future, **we will utilize data contamination detection methods like MM-Detect [7] to more thoroughly investigate the influence of table content contamination on the NIAT benchmark performance. Moreover, constructing NIAT queries based on synthetic tables or tables from more recent table datasets can also alleviate this problem.** A dedicated subsection will be added to incorporate your advice together with related discussions and experiments.
>
> [1] Qwen2.5 Technical Report
>
> [2] OctoThinker: Mid-training Incentivizes Reinforcement Learning Scaling
>
> [3] Understanding R1-Zero-Like Training: A Critical Perspective
>
> [4] DIN-SQL: Decomposed In-Context Learning of Text-to-SQL with Self-Correction
>
> [5] HiTab: A Hierarchical Table Dataset for Question Answering and Natural Language Generation
>
> [6] MultiHiertt: Numerical Reasoning over Multi Hierarchical Tabular and Textual Data
>
> [7] Both Text and Images Leaked! A Systematic Analysis of Multimodal LLM Data Contamination.

---

> > ### Comment · Reviewer_QGzx · 2025-08-04
> >
> > Thank you for the thorough response. The authors have addressed many of my concerns, and I appreciate the clarifications provided.
> >
> > I will raise my score to 4.
> >
> > A few suggestions moving forward:
> > 1) I think including the additional model experiments, contamination analysis, and analysis details as outlined in the rebuttal would strengthen the paper.
> > 2) I believe the paper's core contribution should focus on the clear failure case of locating a cell. Other aspects, such as dataset creation for fine-tuning, can be presented as secondary. Additionally, I would caution against emphasizing long-context challenges too heavily, as failures are evident even without long contexts. That said, including a long-context variant of the benchmark could be useful for future, especially if the current benchmark becomes saturated.

---

> > > ### Author Response · Authors · 2025-08-06
> > > **Response to Reviewer QGzx**
> > >
> > > Thank you for the detailed response and valuable suggestions, which will be fully incorporated in the next version. For additional model experiments, we plan to try our best to add them into the main body if the page limitation permits. For contamination analysis and more analysis details, they could be added into appendix to save space and more obvious pointers will be added in the main body. Moreover, we will adjust the presentation of our paper to focus more on the failure mode of cell-locating task, and further compress other secondary aspects such as fine-tuning data synthesis or move them into appendix.
> > >
> > > Thanks again for replying to us. Please feel free to propose any further questions or concerns if they have not been fully addressed with our current responses.

---

### Official Review · Reviewer_UMxE · 2025-06-19

**Clarity:** 3
**Significance:** 3
**Originality:** 3
**Rating:** 5
**Confidence:** 4

**Summary:**

This paper introduces a new long-context tabular benchmark dataset, named Needle-in-a-table (NIAT), to benchmark language models' performance on extracting the target cell based on location or lookup queries in long tables. It performs a thorough analysis to evaluate various LLMs and MLLMs performance on different table structures and formats. The authors also propose a simple data synthesis method, using strong models to generate reseasoning processes to create the fine-tuning data on the NIAT tasks, demonstrating significant improvement on four downstream tasks, including table question answering and table fact verifications.

**Questions:**

* Maybe I didn't understand it correctly. Could the authors clarify the difference between Multi-Slash and Local-Triangle patterns in Figure 3? It seems to me that in the Local-Triangle pattern, the attention weights are also concentrated on the cell tokens in the same column. Also, it would be clearer to include the legend in Figure 3.
* In Figure 4, for the Llama3.1-8B models trained with the NIAT data, it seems to me there is no significant improvement on per-cell accuracy, unlike Qwen2.5-7B models. There seems to be even a degradation in training using 32x32 cropped data on Llama3.1-8B models. Could the authors comment on the potential reasons? Also in Table 5 in Appendix, the table shows improvement on training Llama3.1-8 B-Instruct models with cell-locating and cell-lookup data. Why is there a difference?
* In Table 3, there is also a degradation on TabFact downstream task on Qwen2.5-7B-Instruct trained using NIAT, could the authors comment on the potential reasons?
* In Table 5, there is a typo in the second row, second column: 17.33 should be in bold for NIAT with 12 table size. Also, it would be more consistent to use 8x8, 12x12 to denote the table size. Additionally, in this table, there seems to be a degradation on the TABMWP dataset with including cell-locating and cell-lookup data. Why is it inconsistent with the results in Table 3, where we see consistent improvement on TABMWP dataset with Llama3.1-8B-Instruct models?

**Ethical Concerns:**

["NO or VERY MINOR ethics concerns only"]

**Final Justification:**

The authors have clarified my confusion on Figure 3 and also corrected the ablation study results during the rebuttal phase.

In general, I believe this is an insightful work to investigate LLM's capabilities in understanding long structured tables. Currently, I haven't seen a lot of work in this area. The proposed benchmark creation is technically sound and quite intuitive. Therefore, I have given the final rating of 5.

**Limitations:**

Yes.

**Paper Formatting Concerns:**

No.

**Quality:**

3

**Strengths And Weaknesses:**

**Strengths**
* Proposes an important benchmark dataset to measure LLM and MLLM's long-context tabular understanding capabilities.
* The proposed NIAT dataset creation is quite scalable and reliable, as it only involves cell locating and cell lookup queries.
* The proposed benchmark dataset covers different table structures, formats and varying sizes.
* Achieves significant improvement gains on downstream tasks using the proposed data synthesis on NIAT tasks.

**Weakness**
* The maximum length in the NIAT dataset is up to 120K, still not close to the token limit in some long-context LLMs. For example, Gemini supports the context window up to 1M tokens.
* The NIAT tasks only involve cell locating and cell lookup, and could involve more complex tasks.

---

> ### Author Rebuttal · Authors · 2025-07-31
>
> We sincerely thank you for the precious effort and time spent in carefully reviewing our paper. The questions will be properly addressed as we reply, and your valuable advice will be fully incorporated when we refine our paper.
>
> **Q1: The NIAT benchmark does not involve more complex tasks, and the prompt length is less than the token limit of some long-context LLMs like 1M tokens.**
>
> **A1:** To provide the community with the first benchmark on the long-context ability of individual cells within structured tables while aiming to save the cost of data collection, we build the NIAT benchmark based on public table datasets and design two table understanding tasks (cell-locating and cell-lookup), which are simple for humans but can provide a basic testbed for LLMs' understanding of table structures and individual table cells. However, as pointed out in the limitation section, we also totally agree that more complex tabular tasks together with larger tables and more hybrid content (like table-text scenario) should be considered to provide a more thorough evaluation of long-context table understanding. For instance, as mentioned in the related work, a concurrent benchmark LongBench-v2 evaluates the high-level reasoning ability of LLMs over long structured tables using 18 question-answering samples of more complex tasks, which we think is really a good start along this direction.
>
> **Q2: The difference between Multi-Slash and Local-Triangle patterns in Figure 3.**
>
> **A2:** We apologize if our descriptions lead to some misunderstanding. The horizontal and vertical coordinates (i.e., (m, n) ) in Figure 3 represents the table cell in the m-th row and n-th column. Note that the input table is serialized into Markdown sequence in a left-to-right and top-to-bottom order. The cell color represents different attention weights, where brighter colors (yellow and green) indicate larger attention weights and darker colors (dark purple) indicate smaller attention weights.
>
> In the Multi-Slash pattern, the attention weights of one cell are concentrated on the preceding cells in the same column, e.g., the cell at (3,3) (3-rd row and 3-rd column) attend to cells at (2,3) and (1,3). In the Local-Triangle pattern, the attention weights of one cells are concentrated on the preceding cells in the same row, e.g., the cell at (3,3) attend to cells at (3,1) and (3,2). We will add these examples and legend to Figure 3 to make it more intuitive.
>
> **Q3: The performance of Llama3.1-8B on cell-locating (on cropped markdown tables) and TABMWP task differs between Figure 4, Table 3 (main body) and Table 5 (ablation study in appendix).**
>
> **A3:** Thank you for this valuable question which helps us find an important reporting error in Table 5 (ablation study) in the appendix, where we mistakenly confused the ablation experiment results on a subset of test samples (current wrong version) with results on all test samples (the intended version). We sincerely apologize for this oversight, and provide the correct ablation results below, which aligns with the trend in Figure 4 and Table 3 in the main body. The Table 5 will be corrected accordingly in the revised manuscript. Thanks again for helping us catch this error.
>
> | **Model** | **8 x 8** | **12 x 12** | **16 x 16** | **20 x 20** | **24 x 24** | **28 x 28** | **32 x 32** | **WTQ** | **TabFact** | **HiTab** | **TABMWP** |
> | --- | --- | --- | --- | --- | --- | --- | --- | --- | --- | --- | --- |
> | *Llama3.1-8B-Instruct* | 16.88 | **13.56** | **12.94** | **8.98** | **6.34** | **7.25** | **3.83** | 49.90 | 62.80 | 26.10 | 54.78 |
> | + Cell-Locating & Cell-Lookup | **20.10** | 12.75 | 10.21 | 5.51 | 4.51 | 4.88 | 2.49 | **67.43** | **78.57** | **49.41** | 66.15 |
> | + Cell-Locating | 19.76 | 13.49 | 12.06 | 7.86 | 5.31 | 6.22 | 3.07 | 67.33 | 67.45 | 33.44 | 70.50 |
> | + Cell-Lookup | 18.85 | 13.13 | 12.15 | 8.36 | 6.22 | 5.83 | 3.21 | 59.00 | 53.50 | 35.00 | 69.44 |
> | + 4 downstream datasets | 16.28 | 11.12 | 11.41 | 6.11 | 5.92 | 5.40 | 2.17 | 64.78 | 78.13 | 48.22 | **81.79** |
>
> **Q4: In the cell-locating task on cropped markdown tables (Figure 4), NIAT fine-tuning benefits Qwen2.5-7B far more than Llama3.1-8B.**
>
> **A4:** This is indeed a curious phenomenon and we present our results honestly. We speculate the potential reasons are two-fold and are related with models' pre-existing capabilities. On the one hand, as noted in the Qwen2.5 technical report [1], Qwen2.5-series underwent specialized post-training on structured tabular data, which may lead to better table structure understanding ability that our CoT-based fine-tuning data could more effectively "activate" and build upon, leading to a significant boost in table structure comprehension. On the other hand, exisiting RL studies [2, 3] have shown that Qwen2.5-series even the base version possesses better reasoning ability, which makes it more easier to incentivize advanced capabilities for complex multi-step tasks, e.g., interpreting the table structure row by row and then pinpointing the answer cell by column index. By contrast, Llama3.1-series needs extra mid-training to achieve the comparable performance.
>
> **Q5: In Table 3 (main body), the performance of Qwen2.5-7B decreases with synthetic NIAT fine-tuning data.**
>
> **A5:**  We speculate a potential reason is that the model may be confused between the GPT-4o generated NIAT fine-tuning data and the learned table fact verification ability during Qwen2.5's post-training, leading to some performance fluctuation, as Qwen2.5 could have used these datasets within post-training stage. Another potential reason is that the NIAT fine-tuned Qwen2.5 does not strictly follow the output format requirements on some test samples, where the evaluation script may wrongly judged them as incorrect samples (i.e., false negative). We will further look into the model output and improve the answer parsing and evaluation scripts if necessary.
>
> **Q6: Typo and other presentation suggestions**
>
> **A6:**  The paper presentation will be revised based on your valuable suggestions. Thanks again for your careful reviewing.
>
> [1] Qwen2.5 Technical Report
>
> [2] OctoThinker: Mid-training Incentivizes Reinforcement Learning Scaling
>
> [3] Understanding R1-Zero-Like Training: A Critical Perspective

---

> ### Comment · Reviewer_UMxE · 2025-08-05
> **Response to authors**
>
> Thank you for the clarifications on two patterns in Figure 3 and for providing the correct ablation study results in Table 5. Please include those changes in the final version.
>
> I will raise my score to 5.

---

### Official Review · Reviewer_E6j3 · 2025-07-02

**Clarity:** 3
**Significance:** 3
**Originality:** 3
**Rating:** 5
**Confidence:** 3

**Summary:**

The paper explores the problem of dealing with long-context table question answering. The authors propose a testing benchmark NEEDLEINATABLE, which covers tables of diverse structures, formats and sizes suited to evaluate models’ underlying fine-grained understanding and perception of individual cells within tabular context.

**Questions:**

- The literature on tabular QA (https://ojs.aaai.org/index.php/AAAI/article/view/34739) and LLM for table understanding (https://link.springer.com/article/10.1007/s11704-024-40763-6) is quite extensive. The related work section is quite concise and not sufficiently detailed. What are the key differences with SOTA?
- A more detailed description of the experiments, parameter settings, and ablation studies is necessary to ensure reproducibility of the research.
- What are the execution times and memory requirements?

**Ethical Concerns:**

["NO or VERY MINOR ethics concerns only"]

**Final Justification:**

The paper contribution is solid and well presented. The empirical evaluations are sufficient to support the initial claims. Therefore, I recommend paper acceptance.

**Limitations:**

- The cost of using proprietary LLMs
- The risks of LLMs' hallucination/bias
- The lack of human-in-the-loop (no relevance feedback)

**Paper Formatting Concerns:**

No concerns (but I did not check it completely)

**Quality:**

3

**Strengths And Weaknesses:**

+ Clear problem formulation and motivations
+ Interesting research direction
+ Promising results

- Incomplete/too high-level related works overview
- Lack of details on reproducibility  (code/experiments/settings)

---

> ### Author Rebuttal · Authors · 2025-07-30
>
> We sincerely thank you for your valuable comments and positive feedback. Your questions will be properly addressed as we reply and your constructive suggestions will be fully incorporated in the next version of our paper.
>
> **Q1: Add more detailed discussion of related work and difference with previous SOTA methods.**
>
> **A1:** Due to the strict page limits of the initial submission, we provide a relatively concise discussion of related works and focus on distinguishing our work from the most recent and relevant lines of research. For instance, we clarify how NIAT is complementary to related long-context benchmarks and tabular benchmarks. Moreover, we totally agree that providing a more detailed context can better highlight our work's unique contributions. A more extensive discussion of related literatures like [1] and [2] will be included in the future version.
>
> When it comes to previous SOTA models like TableGPT2 and QwQ-32B, existing studies primarily evaluate their high-level table understanding ability with complex tasks as testbeds, such as tabular QA and fact verification. By contrast, our constructed NIAT benchmark aims to evaluate the fine-grained understanding of each input table cell from the long-context perspective, making it complementary to previous benchmarks. In addition, our systematic evaluation provides a comprehensive performance landscape of different models including previous SOTA methods, revealing their potential drawbacks towards robust long-context table understanding capacities.
>
> **Q2: Add detailed experimental settings and ablation studies.**
>
> **A2:**  Due to space limitation, we provide important experimental settings in the Section 3.3 and Section 4.2. More detailed experimental settings such as parameter settings and training details were provided in the Appendix C. Ablation studies were also conducted and results were shown in Appendix D.2. Preliminary benchmark data were also included in the supplementary materials. We will revise the main body of the paper to include more key details and add more explicit pointers to the relevant sections in the appendix. Besides, related data, model and code will be open-sourced to further improve reproducibility and facilitate future research in this area.
>
> **Q3: What is the the execution time and memory requirements.**
>
> **A3:** As mentioned in the Appendix C.3, we use 8 NVIDIA A100-80G GPUs for all experiments. For model training, we utilized the Megatron-LM framework for distributed training of Llama3.1-8B and Qwen2.5-7B on 12K synthetic data samples, which took approximately 3 hours on 32 A100 GPUs. For inference, the vLLM framework was employed to accelerate the process for our 287K test samples. The inference time is dependent on model size; for instance, Llama3.1-8B-Instruct achieved a throughput of approximately 700 samples per hour using tensor parallelism on 4 A100 GPUs.
>
> Thank you once again for helping us improve our paper. Above information together with more discussions like the cost of proprietary LLM-APIs and the risk of LLMs' hallucination will be included in the future version.
>
> [1] TableBench: A Comprehensive and Complex Benchmark for Table Question Answering
>
> [2] Large language model for table processing: a survey

---

> > ### Comment · Reviewer_E6j3 · 2025-08-04
> > **Response to authors' feedback**
> >
> > The authors' feedback looks reasonable. I confirm my positive rating on the paper.

---

### Official Review · Reviewer_5fM7 · 2025-07-03

**Clarity:** 3
**Significance:** 4
**Originality:** 3
**Rating:** 5
**Confidence:** 5

**Summary:**

This paper introduces NeedleInATable (NIAT), a novel benchmark for evaluating the long-context understanding capabilities of large language models (LLMs) in structured tabular data. The authors argue that prior long-context benchmarks have focused almost exclusively on unstructured text, overlooking structured tables’ unique challenges—particularly the need for precise structural reasoning and cell-level localization.

NIAT is built upon public table datasets and defines two task types: (1) cell-locating queries that require retrieving the content at a specified row and column, and (2) cell-lookup queries based on simple natural language questions. The authors evaluate a range of LLMs and multimodal LLMs (MLLMs) and reveal large performance gaps between the two query types, highlighting weaknesses in structural understanding. They further propose a data synthesis method using GPT-4o with Chain-of-Thought (CoT) reasoning to fine-tune models, showing improvements on both NIAT and downstream table QA tasks.

**Questions:**

See weakness.

**Ethical Concerns:**

["NO or VERY MINOR ethics concerns only"]

**Final Justification:**

The rebuttal has addressed my concerns and I would like to keep my positive rating.

**Limitations:**

The authors acknowledge that their benchmark uses medium-sized tables and only supports English. However, they do not explicitly discuss the risk of table content overlap with LLM training corpora, or the format-structure mismatch issues. These should be addressed in the limitations section.

**Quality:**

3

**Strengths And Weaknesses:**

### **Strengths**

* **Clear motivation**: The work tackles an under-explored but critical problem—the structural localization of information in long tables—complementing existing long-context benchmarks that focus solely on plain text.
* **Well-constructed benchmark**: NIAT covers different table formats (Markdown, HTML, image) and structures (flat, hierarchical, horizontal), and provides detailed statistics and task definitions.
* **Empirical insight**: The finding that LLMs perform poorly on cell-locating but decently on lookup-style tasks provides valuable diagnostic information for the community.
* **Effective synthetic data strategy**: The CoT-style data generation and fine-tuning approach leads to tangible gains across NIAT and traditional table QA datasets.

### **Weaknesses**

1. **Lack of contamination analysis**: Since the tables are sampled from existing datasets (e.g., WTQ, HiTab), many of them may appear in LLM pretraining corpora, especially for large proprietary models like GPT-4o. While the NIAT questions are novel, the table content may not be. This raises concerns about whether the observed performance (especially improvements on downstream benchmarks) could stem from memorization. A data contamination analysis is essential.

2. **Limited discussion of format appropriateness**: The paper compares Markdown and HTML formats but doesn’t examine their suitability for different table types. For instance, Markdown cannot express hierarchical table structures (e.g., multi-row headers, merged cells), which HTML can. It would strengthen the paper to analyze when one format is preferable and explore alternative formats like JSON or XML that are common in table structure recognition tasks from computer vision.

---

> ### Author Rebuttal · Authors · 2025-07-27
>
> We sincerely thank you for your thorough review and for recognizing our contributions in this work! The questions and problems will be properly addressed as we reply, and your valuable advice will be fully incorporated when we refine our paper.
>
> **Q1: Lack of contamination analysis about used table data.**
>
> **A1**: Thank you for this valuable suggestion. The NIAT queries are constructed based on tables from previous academic datasets, which could indeed have been used in the pre-training stage of some proprietary models like GPT-4o and even some open-source models (as they did not fully open-source their training data). Nevertheless, none of these models achieved a near-perfect performance on the NIAT benchmark especially on the cell-locating task, which indicates that **only memorizing table content during pre-training does not guarantee a robust fine-grained table understanding ability**. In the future, we will utilize data contamination detection methods like MM-Detect [1,2] to more thoroughly investigate the influence of table content contamination on the NIAT benchmark performance. Moreover, constructing NIAT queries based on synthetic tables or tables from more recent table datasets can also alleviate this problem.
>
> As for the performance improvements of our synthetic NIAT training data, our experimental setup was **specifically** designed to investigate "whether enhancing NIAT capabilities can improve models' overall table understanding ability". As a result, **we carefully synthesize fine-tuning samples of NIAT tasks that are different from the test samples in the downstream benchmarks in order to prevent test data memorization.** The experiment results demonstrate that the learned table structural and content interpretation ability with our synthesized NIAT samples can transfer to downstream tasks and thus can improve model performance. **We will use more recent tabular benchmarks with less data leakage risks like RealHiTBench [3] to evaluate model performance and further validate our findings.** In our revised manuscript, we will add a dedicated subsection titled "Discussion on Data Contamination and Memorization" to incorporate your advice together with related discussions and experiments.
>
> **Q2: Discussion about suitable format for different table structures and more alternative table formats.**
>
> **A2**: To save the cost of model inference, we consider three common table formats (Markdown, HTML and image) to control inference sample number. Our experimental results show that models obtain better average performance on Markdown formats for flat and hierarchical tables, but HTML leads to better performance for horizontal tables. We also find that different models can excel at different table formats. Besides, we notice that existing studies have not yet reached an agreement on this issue. For instance, [3] and [4] find that Latex and HTML formats results in the best performance in their experiments, respectively. Thus, we will dedicate more efforts to investigate this problem and evaluate model performance with other alternative table formats like XML and JSON. Related discussions and experiments will be incorporated in the limitations section.
>
> [1] Both Text and Images Leaked! A Systematic Analysis of Multimodal LLM Data Contamination.
>
> [2] Reasoning or Memorization? Unreliable Results of Reinforcement Learning Due to Data Contamination.
>
> [3] RealHiTBench: A Comprehensive Realistic Hierarchical Table Benchmark for Evaluating LLM-Based Table Analysis.
>
> [4] Table Meets LLM: Can Large Language Models Understand Structured Table Data? A Benchmark and Empirical Study

---

> > ### Comment · Reviewer_5fM7 · 2025-08-03
> >
> > Thank you for the thorough rebuttal - it has addressed my concerns and I will keep my relatively high rating.

---

### Note · Authors · 2025-08-14

We extend our sincerest gratitude to the Area Chair and all reviewers (5fM7, E6j3, UMxE, QGzx) for their meticulous reviews and constructive engagement. The discussion period was exceptionally valuable and will significantly improve our final manuscript.

We are particularly grateful for the reviewers' positive reception to our rebuttal, which culminated in score increases from Reviewer UMxE (to 5) and Reviewer QGzx (to 4). Their instrumental feedback enabled us to correct an error in our ablation study (Table 5) and prompted us to conduct additional experiments on latest models to further strengthen our claims.

This process has sharpened our paper's focus on its core contribution: establishing NIAT as a crucial benchmark for the fundamental, yet surprisingly challenging, task of cell localization in large structured tables. The reviewers' feedback has affirmed our conviction that investigating this foundational capability is a vital step towards developing more robust and reliable LLMs for structured data understanding.

We are fully committed to incorporating all suggestions and promised revisions into the next version of manuscript. This will include the new experimental results, a dedicated discussion on data contamination, and enhanced clarifications throughout. To further foster community engagement, we will not only open-source our complete code and data but also establish a public leaderboard for the community to track and compare model performance on the NIAT benchmark.

Thank you once again for your time and thoughtful consideration.

---

### Decision · Program_Chairs · 2025-09-17

**Decision:**

Accept (poster)

**Comment:**

The paper introduces NIAT (NeedleInATable), a new benchmark to evaluate long-context understanding in LLMs and MLLMs on structured tabular data. The authors highlight that existing benchmarks overlook the unique challenges of tables, such as precise structural and cell-level reasoning. NIAT includes two tasks: cell-locating queries (finding a cell by its row/column) and cell-lookup queries (finding a cell based on natural language).

Consensus & Decision
- All reviewers rated the paper positively, finding its motivation and contributions to be significant and novel. They praised the well-constructed benchmark and the empirical finding that LLMs struggle with cell-locating tasks despite performing well on more complex lookup tasks. The proposed data synthesis method for fine-tuning models was also considered effective.
- The primary concerns raised by reviewers—data contamination, limited discussion of table formats, and reproducibility—were all satisfactorily addressed by the authors in their rebuttal. They committed to conducting further analysis, expanding discussions, and open-sourcing their code.
- Based on the paper's clear strengths and the authors' effective responses to all concerns, the AC recommend acceptance. The work provides a critical tool for diagnosing and improving LLMs' ability to handle structured data, a key area of future research.

Suggestions:
- The authors' approach to synthetic data generation should be contextualized within recent related work like "TableRAG: Million-Token Table Understanding with Language Models" (NeurIPS'24) and "Chain-of-Table: Evolving Tables in the Reasoning Chain for Table Understanding" (ICLR'24).